# Genetics re-establish the utility of 2-methylhopanes as cyanobacterial biomarkers before 750 million years ago

Yosuke Hoshino [1,2] ✉, Benjamin J. Nettersheim [3] ✉, David A. Gold [4], Christian Hallmann[1], Galina Vinnichenko [5], Lennart M. van Maldegem[5], Caleb Bishop[5], Jochen J. Brocks [5] & Eric A. Gaucher [2]

Fossilized lipids offer a rare glimpse into ancient ecosystems. 2-Methylhopanes in sedimentary rocks were once used to infer the importance of cyanobacteria as primary producers throughout geological history. However, the discovery of hopanoid C-2 methyltransferase (HpnP) in Alphaproteobacteria led to the downfall of this molecular proxy. In the present study, we re-examined the distribution of HpnP in a new phylogenetic framework including recently proposed candidate phyla and re-interpreted a revised geological record of 2-methylhopanes based on contamination-free samples. We show that HpnP was probably present in the last common ancestor of cyanobacteria, while the gene appeared in Alphaproteobacteria only around 750 million years ago (Ma). A subsequent rise of sedimentary 2-methylhopanes around 600 Ma probably reflects the expansion of Alphaproteobacteria that coincided with the rise of eukaryotic algae—possibly connected by algal dependency on microbially produced vitamin $B_{12}$. Our findings re-establish 2-methylhopanes as cyanobacterial biomarkers before 750 Ma and thus as a potential tool to measure the importance of oxygenic cyanobacteria as primary producers on early Earth. Our study illustrates how genetics can improve the diagnostic value of biomarkers and refine the reconstruction of early ecosystems.

Cyanobacteria played a crucial role in transforming Earth from its initial anoxic state to a modern, oxygenated system, capable of sustaining increasingly complex life[1,2]. Because cyanobacteria were presumably the only relevant group of oxygenic primary producers for much of the Precambrian[3], their fossilized remnants can serve as an indicator for the presence of oxygenic photosynthesis and relative variations in primary production and carbon burial in the geological past. Yet taphonomic alteration and non-quantitative preservation of cyanobacterial microfossils render their use in Precambrian reconstructions complicated[4]. By contrast, lipid molecules from cyanobacterial cellular membranes are less susceptible to taphonomic bias in environments that favour the preservation of organic matter. Thus, fossilized lipids in the form of biomarker hydrocarbons—in this case hopanoids—can provide important semi-quantitative information for reconstructing the ecological importance of cyanobacteria in the deep past[5], thereby framing the environmental context that is crucially needed for better understanding the evolution of increasingly complex life.

[1]GFZ German Research Centre for Geosciences, Potsdam, Germany. [2]Department of Biology, Georgia State University, Atlanta, GA, USA. [3]MARUM Center for Marine Environmental Sciences and Department of Geosciences, University of Bremen, Bremen, Germany. [4]Department of Earth and Planetary Sciences, University of California Davis, Davis, CA, USA. [5]Research School of Earth Sciences, The Australian National University, Canberra, Australian Capital Territory, Australia. ✉e-mail: yhoshino@gfz-potsdam.de; yhoshino06@gmail.com; bnettersheim@marum.de

Hopanoids are triterpenoid lipids predominantly produced by aerobic bacteria and considered functional analogues of eukaryotic sterols[6,7]. In contrast to the overall wide distribution of hopanoids in bacteria, hopanoids methylated at the C-2 position (2-methylhopanoids) have a much narrower taxonomic representation and were once extensively explored as taxon-specific biomarkers for cyanobacteria[8]. A cyanobacterial association was further supported by frequently observed concurrent records of elevated 2-methylhopane abundance and stable nitrogen isotope ratios suggestive of nitrogen fixation[9,10]−cyanobacteria being the major diazotrophs in modern oceans[11]. Although the physiological roles of methylhopanoids are not yet fully understood, protective roles against environmental stress have been suggested[12,13]. Precambrian rocks were found to generally contain higher proportions of 2-methylated hopanoids relative to Phanerozoic sediments, and this observation was interpreted to reflect an elevated proportion of cyanobacteria as Precambrian primary producers[8,14]. Also throughout the Phanerozoic, systematic trends and fluctuations in 2-methylhopane abundances relative to non-methylated regular hopanes (2-methylhopane index; 2-MHI) were widely used as an indicator of environmental stress and cyanobacterial proliferation[9,15]. Discovery of hopanoid C-2 methyltransferase (HpnP), however, called such interpretations into question by revealing the capacity for 2-methylhopanoid biosynthesis in both Cyanobacteria and Alphaproteobacteria[16,17]. It was even suggested that HpnP first evolved within aerobic Alphaproteobacteria[18] and was only later horizontally transferred to Cyanobacteria. These inferences ultimately led to much uncertainty, debate and a questionable utility of ecological proxies involving 2-methylhopanoids.

Here we report on the existence of diagnostic time windows for fossil 2-methylhopanes in the geological past. Numerous new bacterial phyla have recently been found to possess hopanoid-producing genes[19], and this urges a re-evaluation of 2-methylhopanoid distribution in bacteria and their geobiological implications. We re-examined the distribution of HpnP in the bacterial domain and its evolutionary history using up-to-date genomic datasets and an integrated approach towards phylogenetics[20,21]. In addition, by now, we know that pervasive contamination can overprint Precambrian lipid signatures, and doubts have been raised about previously published Precambrian biomarker records[22]. Hence, our molecular data were combined with new analyses of sedimentary 2-methylhopanes to provide an integrated account of 2-methylhopane production through Earth history.

## Results and discussion

### Hopanoid C-2 methyltransferase in bacteria

HpnP is mostly distributed across a subset of bacteria that possess squalene cyclase (SC), which creates the non-methylated hopanoid structure. Nearly all species that possess both the SC and HpnP genes retain only a single copy of each gene. Synteny (gene co-localization) between the SC and HpnP genes was not observed in any phylum, and thus these two genes may be independently inherited. In the current study, SC was found in 31 bacterial phyla, whereas HpnP was found in 12 phyla (Fig. 1a). This contrasts with only four phyla that were previously known to harbour HpnP-containing species[23,24]. Yet, the distribution of SC and HpnP is sporadic in most phyla that contain these two enzymes, and HpnP in particular is concentrated in three phyla: Rokubacteria, Alphaproteobacteria and Cyanobacteria (Fig. 1a,b) (Supplementary Note 1 and Supplementary Tables 1–5 provide more details). Rokubacteria is a recently proposed candidate phylum[25], and its HpnP homologues have not been described thus far. Despite the sporadic occurrence of SC among bacteria, a recent comprehensive phylogenetic study suggests that SC was present in the individual common ancestors of three HpnP-containing phyla (Rokubacteria, Alphaproteobacteria and Cyanobacteria). This implies multiple events of SC gene loss within individual phyla, and HpnP seems to have followed a similar complex evolutionary history. It is currently not clear what selective pressure caused the loss or retention of the SC and HpnP genes.

### HpnP phylogenetic analysis

The evolutionary relationship of HpnP in 12 bacterial phyla was examined through phylogenetic analyses (Fig. 2 and Supplementary Figs. 1–3). HpnP forms a monophyletic group among multiple clades of uncharacterized HpnP-like proteins (HpnP clade; Fig. 2, insert). HpnP and HpnP-like proteins broadly retain three conserved domains: a B-12 binding domain, a radical SAM domain and a domain of unknown function (DUF4070). Vitamin $B_{12}$ (cobalamin) dependency of HpnP was recently experimentally confirmed[24], and our results suggest this cobalamin dependency is a general characteristic of HpnP homologues. Other distant homologues with a radical SAM domain do not retain all three conserved domains. Currently, methyltransferase activity is confirmed only for several proteins in the HpnP clade, and thus it is unknown if HpnP-like proteins are even involved in hopanoid (or more generally terpenoid) biosynthesis. Individual clades of HpnP and HpnP-like proteins display different taxonomical distributions and the tree topology within those clades generally does not follow the species tree at the phylum level. Some species were found to possess multiple HpnP homologies, in addition to HpnP. For instance, many cyanobacteria possess two HpnP homologues−HpnP and an uncharacterized HpnP-like protein. The latter was once thought to be HpnP[26], but this was later corrected[24]. Those HpnP-like proteins serve as outgroup sequences in the present study.

The deepest-branching HpnP homologues in the HpnP clade were obtained from Deltaproteobacteria (myxobacteria, in particular) that do not possess SC in their genomes (Fig. 2). This suggests that deltaproteobacterial HpnP homologues are not involved in hopanoid biosynthesis (Supplementary Note 2). We cannot exclude the possibility, however, that deltaproteobacteria containing HpnP homologues may acquire and methylate non-methylated hopanoid precursors from the environment. Except for Deltaproteobacteria, most bacteria that possess an early-diverging HpnP homologue also possess SC, and those bacteria presumably (although not experimentally confirmed) have the ability to produce 2-methylhopanoids. Currently, 2-methylhopanoid production has been confirmed only for the three late-branching phyla: Alphaproteobacteria, Cyanobacteria and Acidobacteria[8,27,28]. Thus, characterization of early-diverging HpnP homologues from other bacterial phyla, including Deltaproteobacteria, may provide important information about the functional evolution of radical SAM methyltransferases towards HpnP.

Focusing on HpnP proteins accompanied by SC, the tree topology does not replicate the known relationships between bacterial phyla and a complex history of horizontal gene transfer (HGT) is inferred (Fig. 2). However, Cyanobacteria, Rokubacteria and Alphaproteobacteria are well separated with robust nodal support by both maximum likelihood and Bayesian inferences (Fig. 2 and Supplementary Figs. 1 and 2), and no sign of *recent* HGT is observed. This suggests that the HGT events at the root of those sub-clades are ancient and the bacterial lineages that mediated those transfers (but were not necessarily producing 2-methylhopaonids themselves) are long extinct. Whereas HpnP sequences from Alphaproteobacteria and Cyanobacteria dominate our dataset, the evolutionary pattern seen in the two clades is distinct. Alphaproteobacterial sequences come from a sporadic number of sub-clades, suggesting HpnP emerged late in the group, while cyanobacterial sequences largely mirror the species tree, suggesting HpnP was inherited vertically in crown-group cyanobacteria. These observed patterns are described in more detail below.

### Evolution of HpnP in Alphaproteobacteria

Unlike in previous studies that were based on more limited datasets[16,18], alphaproteobacterial HpnP homologues do not form a monophyletic clade but instead cluster together with homologues from Acidobacteria and the newly proposed candidate phylum Eremiobacterota (Fig. 2). Hence, it is not clear if HpnP appeared only once in Alphaproteobacteria and was later horizontally transferred to the other phyla or if

**Fig. 1 | Distribution of SC and HpnP in bacteria. a**, Distribution in individual bacterial phyla; only phyla that harbour SC are shown. Light yellow colour indicates that only a small number of species possess the gene (below 10% of available genomes). Note that the orange colour does not necessarily mean the gene is ubiquitous in a phylum. Individual proteobacterial classes are treated as phyla in the main text. Abbreviations: FCB, Fibrobacteres–Chlorobi–Bacteroidetes group;

PVC, Planctomycetes–Verrucomicrobia–Chlamydiae group. **b**, Distribution of SC and HpnP within the phyla Cyanobacteria and Alphaproteobacteria. The number of families within individual orders is shown in the second column, while the number of families that contain SC and HpnP is shown in the third and the fourth columns, respectively. Supplementary Tables 1–5 provide the details within individual families.

**b**

**Cyanobacteria**

| Order | Family | SC | HpnP |
|---|---|---|---|
| Chroococcidiopsidales | 1 | 0 | 0 |
| Gloeobacterales | 1 | 1 | 1 |
| Gloeomargaritales | 1 | 1 | 1 |
| Nostocales | 10 | 10 | 10 |
| Chroococcales | 5 | 4 | 4 |
| Oscillatoriales | 6 | 5 | 5 |
| Pleurocapsales | 4 | 3 | 0 |
| Spirulinales | 1 | 0 | 0 |
| Synechococcales | 9 | 6 | 6 |
| Total | 38 | 30 | 27 |

**Alphaproteobacteria**

| Order | Family | SC | HpnP |
|---|---|---|---|
| Caulobacterales | 1 | 0 | 0 |
| Emcibacterales | 1 | 0 | 0 |
| Holosporales | 3 | 0 | 0 |
| Hyphomicrobiales | 17 | 12 | 7 |
| Iodidimonadales | 1 | 0 | 0 |
| Kordiimonadales | 1 | 1 | 0 |
| Magnetococcales | 2 | 0 | 0 |
| Maricaulales | 2 | 0 | 0 |
| Micropepsales | 1 | 0 | 0 |
| Minwuiales | 1 | 1 | 0 |
| Parvularculales | 1 | 1 | 0 |
| Pelagibacterales | 1 | 0 | 0 |
| Rhodobacterales | 3 | 1 | 1 |
| Rhodospirillales | 11 | 9 | 1 |
| Rhodothalassiales | 1 | 1 | 0 |
| Rickettsiales | 4 | 1 | 0 |
| Sneathiellales | 1 | 0 | 0 |
| Sphingomonadales | 4 | 4 | 1 |
| Total | 56 | 31 | 10 |

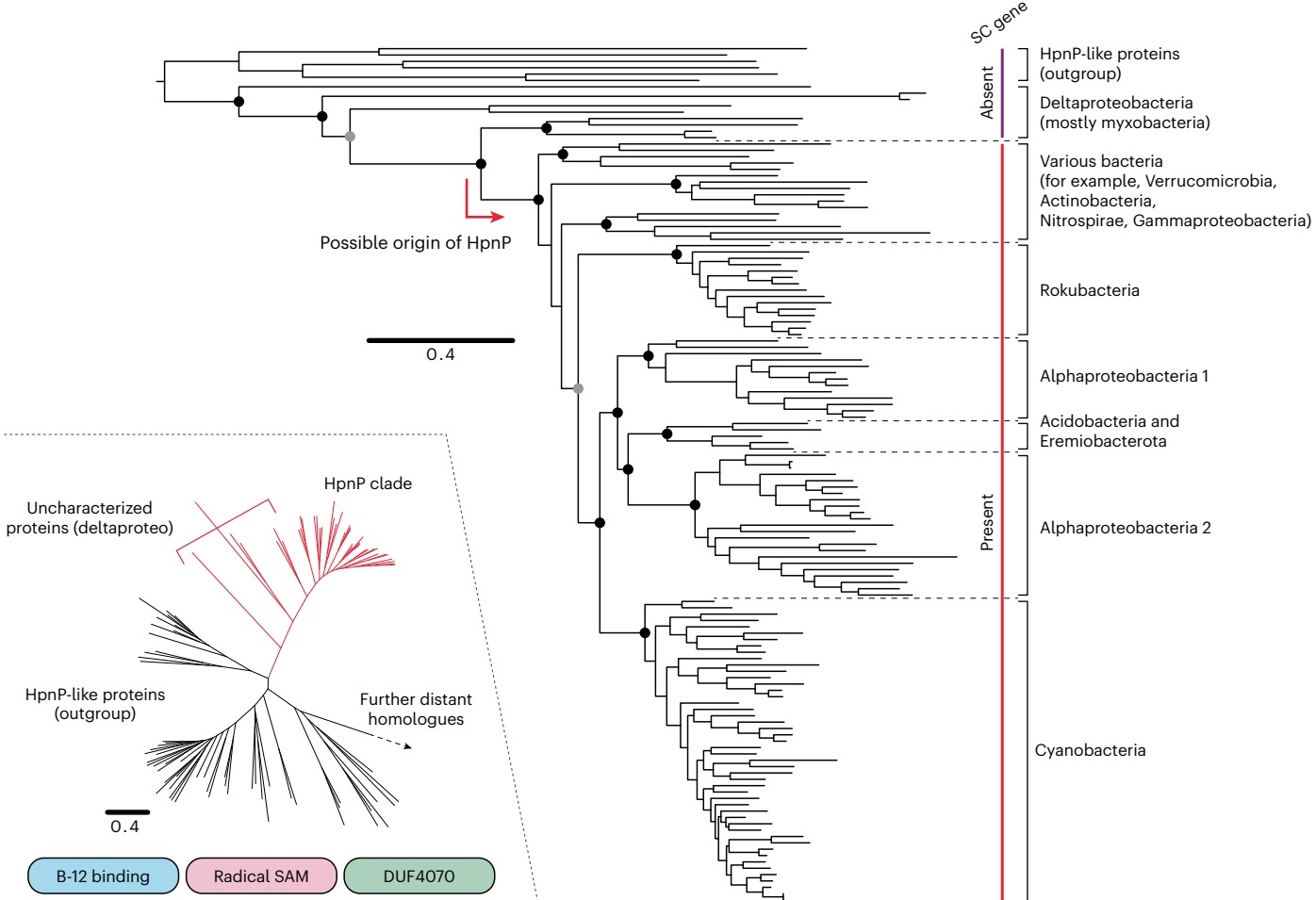

**Fig. 2 | Maximum likelihood tree of HpnP protein sequences.** The tree was generated using 135 representative sequences and 534 conserved sites. Filled black circles indicate that the node support is >85% for both maximum likelihood inference and Bayesian inference (node support is shown only for major clades). Filled grey circles indicate that the node support is above the same threshold for one of the two inferences. The presence/absence of the SC gene is indicated next to the tree. The insert displays major clades of HpnP homologues (HpnP and HpnP-like proteins) that retain three conserved domains; B-12 binding domain, radical SAM domain and DUF4070 domain. The scale bar represents 0.4 amino acid replacements per site per unit evolutionary time. Supplementary Figs. 1–3 provide the complete trees with the species annotation.

Alphaproteobacteria received HpnP through multiple HGT events from different sources. The two alphaproteobacterial clades are taxonomically distinct. One clade mostly consists of the family Methylobacteriaceae and some unclassified alphaproteobacteria, while the other principally contains Beijerinckiaceae and Nitrobacteraceae (Extended Data Fig. 1). These three alphaproteobacterial families belong to the same order Hyphomicrobiales (formerly Rhizobiales), while HpnP is nearly absent in other families (17 families in total; Supplementary Tables 2 and 3). Further, the observed HpnP tree topology is not consistent with the currently inferred species tree of Hyphomicrobiales (Extended Data Fig. 1), and HpnP is nearly absent from other alphaproteobacterial orders (Fig. 1b). Thus, we infer that HpnP was horizontally transferred to Alphaproteobacteria only after the divergence of Hyphomicrobiales, possibly multiple times independently.

### Evolution of HpnP in Cyanobacteria

In contrast to Alphaproteobacteria, the HpnP branching order in Cyanobacteria broadly matches the species tree, despite the overall distribution of HpnP being sporadic at the species level (Fig. 3 and Supplementary Table 4). For instance, HpnP is found in two early-branching taxa (*Gloeobacter* and *Gloeomargarita*), and HpnP homologues from these taxa also branch early in the cyanobacterial HpnP tree. Here we examined the evolutionary history of cyanobacterial HpnP through reconciliation analyses of its tree topology. The reconciliation between the cyanobacterial HpnP tree and the species tree suggests that early-branching orders of the HpnP tree are well consistent with the species tree, while multiple transfer events of the HpnP gene seem to have occurred in individual sub-clades (Fig. 3 and Supplementary Figs. 4 and 5). The frequency of HGT is estimated to be no higher than ~15% (13 out of 86 branches) within HpnP-containing cyanobacterial species. Some putative transfer events are likely to be artefacts due to an inaccurate tree topology with low node supports and the heterotachy of some sub-clades in the HpnP tree (for example, weak node supports for the lineage that splits from *Neosynechococcus*; Fig. 3). The absence of HpnP in the majority of cyanobacteria is attributed to gene loss that seems to have occurred multiple times independently in major lineages (Extended Data Fig. 2 and Supplementary Note 3 for detailed discussions). Our results suggest that HpnP was present in the last common ancestor of Cyanobacteria and was subsequently vertically inherited, with multiple gene loss events and potentially some transfers within the phylum. SC has similarly been inferred to have been present in the last common ancestor of Cyanobacteria and vertically inherited[19]. However, this does not necessarily mean that SC and HpnP co-evolved,

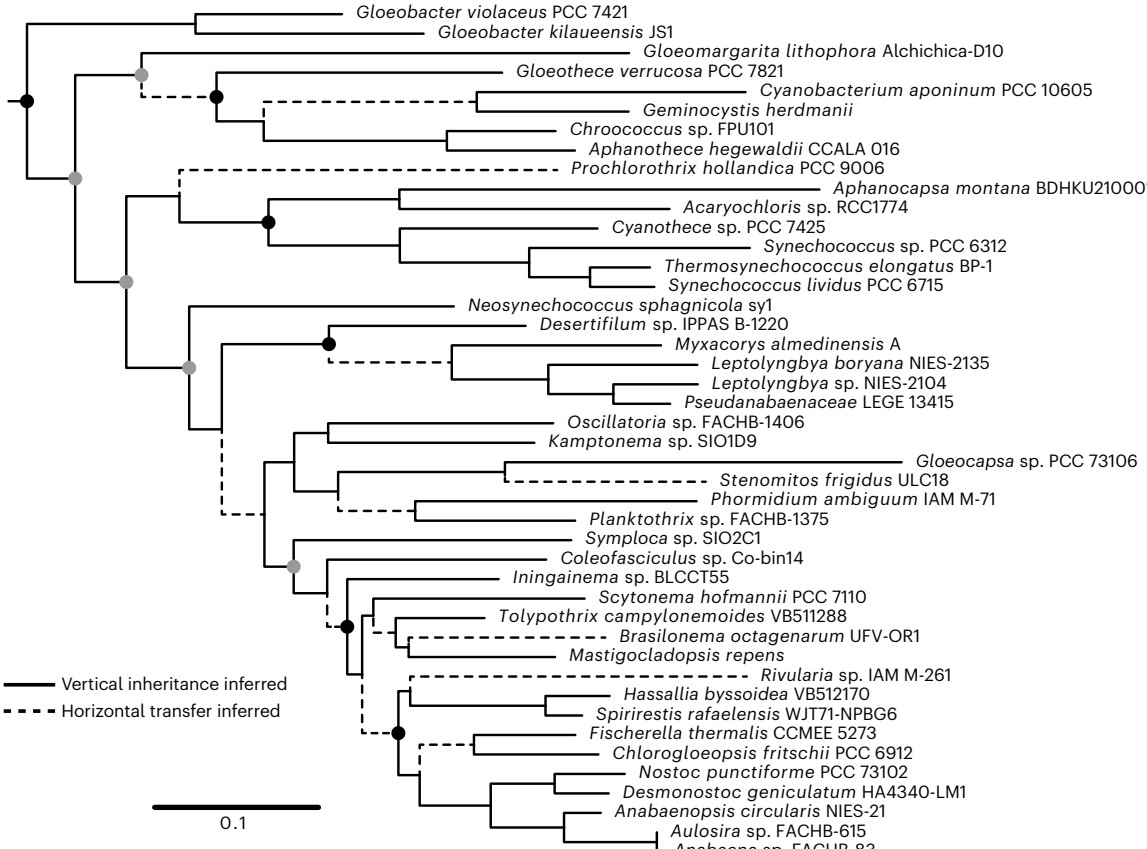

**Fig. 3 | Maximum likelihood tree of HpnP in Cyanobacteria.** A species tree that contained 44 representative cyanobacterial species (Supplementary Fig. 4) was compared to the HpnP tree through reconciliation analyses (Supplementary Fig. 5 for the comparison result and HGT inference by the reconciliation analysis software Notung). Solid lines in the HpnP tree indicate that they are consistent with the species tree. Dashed lines indicate that they are not consistent with the species tree, and thus the presence of HGT is inferred by Notung. Filled black circles indicate that the node support is >85% for both maximum likelihood inference and Bayesian inference (node support is shown only for major clades). Filled grey circles indicate that the node support is above the same threshold for one of the two inferences. The scale bar represents 0.1 amino acid replacements per site per unit evolutionary time.

because no synteny is observed between the SC and HpnP genes. The ability to produce C-2 methylated hopanoids was thus probably present in the common ancestor of crown-group cyanobacteria, which contrasts with the here inferred late origin of alphaproteobacterial 2-methylhopanoid production (Supplementary Note 4 provides comparison with previous studies).

## 2-Methylhopanoids as aerobiosis biomarkers

The distribution of HpnP is nearly exclusive to obligate and facultative aerobes, even though the HpnP catalysis itself does not require molecular oxygen. HpnP-containing species constitute a subset of SC-containing bacteria, which are also mostly aerobes, although SC is additionally found in several anaerobic lineages (for example, Brocadiales, Desulfovibrionales)[29,30]. Hence, fossilized 2-methylhopanoids (2-methylhopanes) in the geological record are probable markers for aerobic bacteria and thus aerobic environments. This is consistent with an environmental survey that associated HpnP with microaerobic environments[17]. HpnP most likely evolved in, or was acquired by, stem-group cyanobacteria after the split from non-photosynthetic lineages (for example, Melainabacteria, Margulisbacteria; HpnP absent). In this framework, the analysis of geological 2-methylhopane records as aerobiosis markers potentially enables us to address the timing of incipient oxygen production before the onset of the GOE at ~2.4 billion years ago (Ga) as was originally proposed[8], regardless of the phylogenetic identity of the biological host, if thermally well-preserved sedimentary sequences should ever be found[22].

## Source of fossilized 2-methylhopanoids

Understanding the evolutionary timeline of HpnP in individual phyla, in particular Cyanobacteria, Alphaproteobacteria and Rokubacteria, allows constraining the source of 2-methylhopanoids at different geological times. HpnP in additional bacteria suggests that Cyanobacteria was not the only lineage capable of producing 2-methylhopanoids. However, HpnP appears only sporadically in those additional lineages, possibly reflecting relatively recent HGT events or limited microbial sampling. Similarly, the presence of these proteins at the base of the HpnP tree could represent long branch attraction of proteins that are dissimilar from the better sampled clades (Cyanobacteria and Alphaproteobacteria)—perhaps representing a divergent function—or they could represent ancestry from an extinct lineage with an ancestral form of HpnP that was not necessarily involved in 2-methylhopanoid production, as implied for deltaproteobacterial HpnP homologues. We currently do not have enough data to adjudicate between these competing interpretations.

Nevertheless, accepting these limitations, we can conclude that 2-methylhopanoid biosynthesis existed in Cyanobacteria before it appeared within Alphaproteobacteria. The evolutionary history of Rokubacteria is not constrained, and it remains unclear if any members actually produce geologically relevant amounts of 2-methylhopanoids. Rokubacteria are neither primary producers, nor have they been detected in marine settings. Also, rokubacterial HpnP seems confined to several late-branching clades, although the exact timing of HpnP acquisition is not clear (Extended Data Fig. 3 and Supplementary Figs.

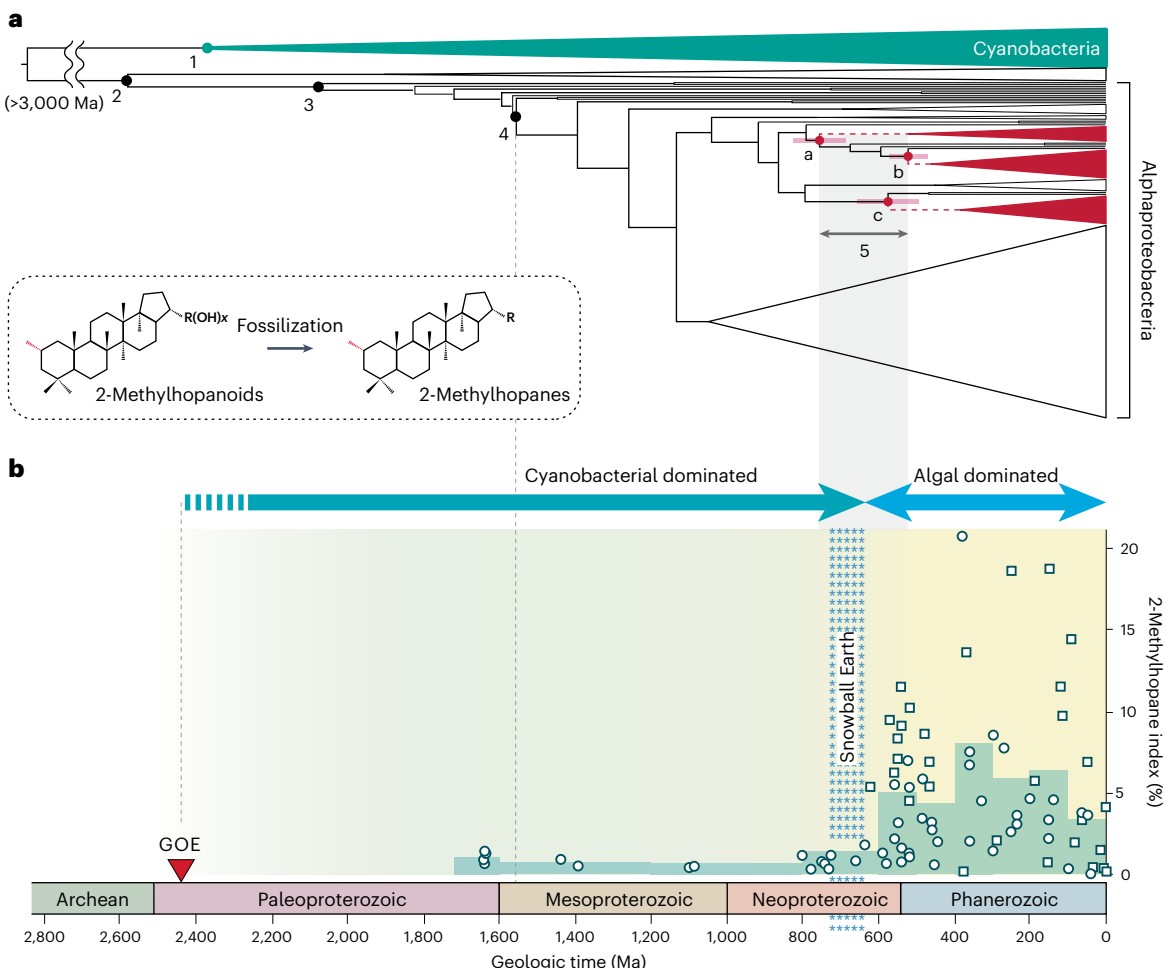

**Fig. 4 | Evolutionary timeline of HpnP and geological record of 2-methylhopanes throughout Earth's history. a**, Dated phylogeny of Alphaproteobacteria and the emergence of HpnP in the phylum. Both the tree topology and the node dates were adapted from a recent phylogenomic study (Supplementary Fig. 10)[32]. Numbers in the tree indicate major evolutionary events for the two phyla Cyanobacteria and Alphaproteobacteria: (1) divergence of crown-group cyanobacteria; (2) divergence of Alphaproteobacteria from Beta- and Gammaproteobacteria; (3) divergence of crown-group Alphaproteobacteria; (4) divergence of Hyphomicrobiales; (5) divergence of HpnP-containing Hyphomicrobiales families—Beijerinckiaceae (node a), Methylobacteriaceae (node b) and Nitrobacteraceae (node c). Nodes a, b and c indicate the emergence of three HpnP-containing families in Hyphomicrobiales. Node bars denote the 95% highest posterior density interval of posterior dates. Supplementary Fig. 10

provides the dated phylogeny with the species annotation. **b**, 2-Methylhopane index (2-MHI, % $C_{31}$ 2-methylhopane / ($C_{31}$ 2-methylhopane + $C_{30}$ αβ-hopane)) throughout Earth's history. The data contain newly analysed values from this study (open circles) and from the literature (open squares). Each circle and square represent the average of all 2-MHI values from individual geological formations or published studies (Supplementary Table 7). The underlying pale green bar diagram provides averages binned for geological time units (Supplementary Table 7). Blue-green and light-blue arrows indicate two geological periods when marine primary productivity was dominated by Cyanobacteria and algae, respectively. Abbreviations: GOE, Great Oxidation Event; Ma, million years ago. Figure adapted from ref. 32 under a Creative Commons license CC BY 4.0.

7–9). Hence, it appears unlikely that Rokubacteria substantially contributed to the Precambrian marine biomarker record. The same applies to other sporadic occurrences of HpnP discussed above. In contrast, cyanobacteria are generally accepted as important primary producers that substantially contributed to organic carbon burial throughout the mid-Proterozoic[3,31]. Also, our genome data reveal that HpnP is in fact distributed much more widely in marine cyanobacteria than previously known[23] (Supplementary Table 5 and Supplementary Note 5 for more details). Therefore, cyanobacteria are the most likely source of fossilized 2-methylhopanoids before the evolution of HpnP-containing Alphaproteobacteria (Hyphomicrobiales).

Constraining the emergence of HpnP-containing Hyphomicrobiales then provides a means to determine geological periods and formations with (nearly) exclusively cyanobacterial signatures. The evolutionary timeline of HpnP in Alphaproteobacteria was estimated using recently published phylogenomic analyses of

Hyphomicrobiales (Fig. 4a)[32]. This published dated phylogeny suggests that Hyphomicrobiales emerged around 1.5 Ga (node number 4 in Fig. 4a), while the emergence of the stem groups of three major HpnP-containing families occurred only after 750 Ma (number 5 in Fig. 4a; nodes a, b and c). The phylogenetic relationship between those families in the HpnP tree is not consistent with the species tree of Hyphomicrobiales, as described above (Extended Data Fig. 1). Thus, the HpnP gene was probably horizontally transferred to those three families after 750 Ma. In contrast, HpnP is widespread within the individual three families, suggesting the presence of HpnP in the common ancestors of each of those individual families. The divergence of those three families is estimated to have occurred after 500 Ma (Fig. 4a; three triangles in red)[32]. Therefore, the emergence of HpnP in Alphaproteobacteria can be estimated to having occurred between the emergence of the stem groups and the divergence of the crown groups of three HpnP-containing families, somewhere

between 750 and 500 Ma (node number 5; Fig. 4a). In turn, fossilized 2-methylhopanoids in marine sedimentary rocks deposited before 750 Ma probably indicate cyanobacterial sources.

### Non-contaminated 2-methylhopane record in the Precambrian

The record of the 2-methylhopane index (2-MHI) published thus far is generally characterized by high values in Precambrian rocks and oils (up to 19%), relative to those in younger organic matter, and an elevated proportion of cyanobacterial primary production was thus inferred for the early Earth[8,14]. However, the syngeneity (analytes having the same age as the host rock) of 2-methylhopanes in Precambrian samples has never been adequately and systematically assessed. By now, it is widely recognized that even traces of contamination can pose a serious problem for organic-lean Precambrian samples, as was previously scrutinized for Archaean biomarker studies[22]. Hence, we re-investigated Precambrian rocks using rigorous slice-extraction and interior–exterior experiments, allowing us to separately study non-contaminated sample interiors and contamination-prone sample exteriors (surfaces)[22,33]. Our analyses revealed that high 2-MHI values are restricted to sample exteriors—up to 17% in the 1.64 Ga Barney Creek Formation that hosts the oldest known 2-MHI record[33] (Supplementary Note 6 and Supplementary Table 6)—whereas non-contaminated interiors exhibit 2-MHI values with an average of 1.1% (n = 42) and a maximum of only 2.4% (Extended Data Fig. 4). Abiotic hopanoid methylation at the position C-2 is highly unlikely (Supplementary Note 7 and Supplementary Fig. 12). As our results highlight the need for a systematic re-evaluation of Precambrian 2-methylhopane abundances, we compiled a new non-contaminated 2-MHI record based on newly analysed rock samples from the Phanerozoic to the Paleoproterozoic that underwent syngeneity assessment (n = 161), combined with previously published 2-MHI values, excluding any potentially contaminated samples from the pre-Ediacaran period (n = 507) (Supplementary Table 7).

In contrast to previous findings, the revised 2-MHI record exhibits consistently low relative 2-methylhopane abundances until the end of the Snowball Earth glaciations around 635 Ma (2-MHI < 2.4%; n = 93; average = 0.96%) (Fig. 4b). An initial increase to 5.4% is observed in the early Ediacaran (~620 Ma) and followed by multiple episodes of large fluctuations. While the 2-MHI in the Ediacaran and Phanerozoic can be extremely high for individual formations (up to 24%), the averaged values over geological time units remain moderate at ~5%.

### Ecological implications of 2-methylhopanes

Comparatively low 2-MHI values (~1%) in the mid-Proterozoic represent the average contribution of cyanobacterial 2-methylhopanoids to mid-Proterozoic sediments and provide a baseline for gauging the activity of cyanobacteria, even though the 2-methylhopanoid production by marine cyanobacteria seems to be confined to nearshore environments[34,35] (Supplementary Table 5 and Supplementary Note 8 for detailed discussions). The 'rise of algae' in the Cryogenian that broke the billion-year incumbency of cyanobacteria was attributed to a changing nutrient regime, which was triggered by massive glaciogenic inputs of phosphorus into oceans and allowed eukaryotic algae to permanently outcompete cyanobacteria[3]. Ecological shifts throughout the Ediacaran period—diminished abundances of microbial mats[36,37] and the expansion of HpnP-free cyanobacteria as major primary producers in pelagic regions (Extended Data Fig. 2)—should have curtailed net cyanobacterial 2-methylhopanoid productions and preservations. This draws particular attention to the unexpected Ediacaran increase of the 2-MHI that probably reflects the emergence of non-cyanobacterial 2-methylhopanoid sources that may have played an important ecological role in Ediacaran oceans (Fig. 4b). Combining the biomarker record with time-calibrated genetic analyses points to the emergence of this lipid biosynthetic capacity in Alphaproteobacteria as an explanation (500–750 Ma; Fig. 4a). This inference is consistent with recent studies

that proposed Alphaproteobacteria as the major source of Phanerozoic 2-methylhopanes[23,24] and also the dominance of alphaproterobacterial HpnP genes in many modern marine metagenomes[17] (Supplementary Notes 4 and 5). Hence, it is likely that the Ediacaran increase of the 2-MHI reflects the expansion of HpnP-containing Alphaproteobacteria.

We hypothesize that this dynamic shift in 2-methylhopanoid producers and the rise of algae proceeded in concert. It is well known that eukaryotes depend on symbiotic microbes for nutrient acquisition and have co-evolved with those symbionts, as observed for algae, plants and animals[38,39]. For instance, eukaryotes cannot biosynthesize vitamin $B_{12}$ (cobalamin) de novo, although the majority of green algae require this nutrient for the biosynthesis of the essential amino acid methionine (Supplementary Note 9)[40]. Dissolved oceanic vitamin $B_{12}$ concentrations are generally too low to sustain the growth of algae, which thus rely on the uptake of microbially produced vitamin $B_{12}$ (ref. 40). Vitamin $B_{12}$ dependency evolved in several algal lineages independently during the course of evolution, through developing mutualistic relationships with microbial donors and retaining an efficient vitamin $B_{12}$-dependent methionine synthase[40]. In modern environments, dominant vitamin $B_{12}$ contributors to algae are Alphaproteobacteria and Gammaproteobacteria[41,42] that utilize an oxygen-dependent pathway for vitamin $B_{12}$ biosynthesis[41], whose presence is predicted for many HpnP-containing Alphaproteobacteria (Supplementary Table 3). Hence, an expansion of vitamin $B_{12}$-producing Alphaproteobacteria in mutualistic relationship with algae during the Ediacaran may be one factor tying the steep rise of 2-MHI values to the rise of algae (Supplementary Note 9 for more details). The elevated 2-MHI may additionally reflect an increased reworking of newly available algal biomass in increasingly oxygenated marine environments by heterotrophic Alphaproteobacteria. These hypotheses are complementary to the previously proposed relationship between 2-methylhopanoid production and the nitrogen cycle during Cretaceous oceanic anoxic events (Supplementary Note 4)[23,24]. In view of enhanced alphaproteobacterial 2-methylhopanoid production going back to the Ediacaran, it is likely that HpnP-containing Alphaproteobacteria were responsible for modulating the 2-MHI throughout the Phanerozoic.

## Conclusions

The once suggested association of 2-methylhopanoids with cyanobacteria—a highly important proxy tool for our understanding of Earth system evolution and oxygen dynamics in the geological past—has been constantly debated. Both Cyanobacteria and Alphaproteobacteria can biosynthesize 2-methylhopanoids, and thus sedimentary 2-methylhopanes are not a diagnostic biomarker for a single taxonomic group. We show that such ambiguities can be refined by adding a genetic perspective. The phylogenetic tree topology of HpnP in Cyanobacteria suggests that HpnP was probably present in the common ancestor of crown-group cyanobacteria, whereas according to molecular clock analyses alphaproteobacterial lineages acquired HpnP after 750 Ma. This finding re-establishes the utility of 2-methylhopanes as a biomarker for cyanobacteria in pre-Ediacaran rocks, enabling us to measure the importance of HpnP-containing oxygenic cyanobacteria in the geological past. By contrast, 2-methylhopanes in post-Cryogenian oceans reflect additional signals from heterotrophic 2-methylhopanoid producers—Alphaproteobacteria. The synchronization between the 2-MHI increase during the Ediacaran and the ecological expansion of HpnP-containing Alphaproteobacteria and eukaryotic algae may not be coincidental, involving a vitamin $B_{12}$-based mutualistic relationship between Alphaproteobacteria and algae and enhanced reworking of algal biomass by Alphaproteobacteria in increasingly oxygenated marine environments. Our study demonstrates the strength of combining the geological record of fossil hydrocarbon biomarkers with genetic analyses to gain insights into ancient ecosystems and provides an important precedent for refining our understanding of biomarker utility throughout Earth's history.

## Methods

### Bioinformatics analyses

**HpnP dataset construction.** Representative sequences for hopanoid biosynthesis enzymes were identified from UniProt (https://www.uni-prot.org/); HpnP (B3QHD1), SC (P33247). These protein sequences were utilized as seeds to identify homologous sequences in all three domains of life. Sequences were retrieved from GenBank (http://www.ncbi.nlm.nih.gov/), using BLASTp and PSI-Blast[43,44], with the cut-off threshold of $<1 \times 10^{-5}$. Taxonomically redundant sequences were excluded. Protein domain identification was performed using HMMER v3.3.1 (ref. 45) with the PFAM-A database (http://ftp.ebi.ac.uk/pub/databases/Pfam/current_release/Pfam-A.hmm.gz).

**Phylogenetic analysis.** Sequences were aligned using Muscle v3.8.31 (ref. 46). Phylogenetic trees were constructed by maximum likelihood inference using IQ-TREE v2.1.06 (ref. 47) and by Bayesian inference using MrBayes v3.2.6 (ref. 48). Substitution models were selected using ModelFinder in IQ-TREE—LG matrix with empirical frequencies estimated from the data ($F$) and the FreeRate model for rate heterogeneity across sites ($R$). For maximum likelihood inference, branch support was obtained by Ultrafast bootstrap in IQ-TREE. For Bayesian inference, two Markov chain Monte Carlo chains were run for at least 1,000,000 generations until the maximum discrepancy between chains was <0.05. A consensus tree was generated from two chains using a burn-in of 10% of sampled points. In initial analyses, several protein sequences were found to display high sensitivity to sequence alignments and instable positioning in generated trees (accession numbers PYX91820 and HBG07613). These sequences were excluded from the subsequent analyses.

**Cyanobacterial species tree construction and gene tree/species tree reconciliation.** The list of cyanobacterial species with HpnP proteins was used to create a species tree. The species names were converted into taxon IDs using the NCBI Taxonomy name/id Status Report Page (https://www.ncbi.nlm.nih.gov/Taxonomy/TaxIdentifier/tax_identifier.cgi). These IDs were used with the research command line utility[49] to download all proteins for each taxon in NCBI's Identical Protein Group (IGP) database. Single-copy orthologous proteins were identified in these IGP datasets using OrthoFinder v2.5.4 (ref. 50). OrthoFinder identified 34 single-copy orthologue groups, which were individually aligned using MUSCLE[46] and then concatenated together into a supermatrix using FASconCAT-G[51]. The species tree was generated from this supermatrix using IQ-TREE, including details of each protein's beginning and end in the supermatrix alignment. We allowed IQ-TREE to determine the best model of amino acid evolution for each protein in the supermatrix individually before performing maximum likelihood inference. Once the species tree was generated, we compared how the topology of the cyanobacterial HpnP gene tree matched with the species tree using NOTUNG v2.9 tree reconciliation (DTL model)[52].

**Rokubacteria species tree construction and gene tree/species tree reconciliation.** The phylum currently contains only metagenomic samples, and thus the same species tree construction as performed for cyanobacteria was not possible. Instead, a species tree was constructed by concatenating three ribosomal proteins L2, L3 and L4 (Supplementary Table 1), using IQ-TREE. The best substitution model was also determined by IQ-TREE. The topology of the rokubacterial HpnP tree was compared with the generated species tree using NOTUNG v2.9 (DTL model).

### Biomarker analyses

**Selection of previous biomarker studies.** Previous studies that are included in our dataset are summarized in Supplementary Table 7 (black colour). Phanerozoic samples are generally organic rich, and thus the influence of trace contamination overprint is probably negligible in most cases. Moreover, a much higher density of available datapoints minimizes the risk of interpreting outliers and unrepresentative samples. Therefore, samples were selected to uniformly cover the whole Phanerozoic period without considering the syngeneity of observed biomarkers. In contrast, Precambrian samples are mostly organic lean and thus are easily overprinted by trace contamination as shown for samples from the 1.64 Ga McArthur Basin (Supplementary Note 6). To minimize uncertainty due to contamination, Precambrian samples in our dataset consist mostly of our newly analysed samples, except for Ediacaran oil samples that are highly organic rich and difficult to substantially adulterate through contamination overprint.

**Biomarker extraction and gas chromatography-mass spectroscopy measurement.** Samples that were newly analysed in this study are summarized in Supplementary Table 7 (highlighted in red). Biomarkers were extracted using interior–exterior and slice-extraction experiments as previously described[22,53]. For the interior–exterior experiments, rock samples were cut into two portions using a diamond saw—the exterior (exposed surface; approximately 3 mm thickness) and the interior (inner core). Slice-extraction experiments are an extension of the interior–exterior experiments, and rocks were cut into more than two slices to analyse the millimetre-scale distribution of biomarkers. Each slice was crushed independently with a stainless-steel puck mill and bitumen extracted with a Dionex Accelerated Solvent Extractor (ASE 200) using dichloromethane (DCM) for GR7 samples[33] and 90:10% DCM:methanol for LV09001 samples. Saturated fractions were eluted with 1.5 dead volumes $n$-hexane on a microcolumn of annealed (300 °C, 12 h) and dry-packed silica gel. D4-$C_{29}$ $\alpha\alpha\alpha$R-ethylcholestane (D4; Chiron Laboratories AS) was added to the saturate fraction before gas chromatography-mass spectroscopy (GC–MS) analyses with an Agilent 6890 gas chromatograph equipped with a 60 m DB-5 MS capillary column (0.25 mm i.d., 0.25 μm film thickness; Agilent Technologies) coupled to a Micromass Autospec Premier double sector mass spectrometer (Waters Corporation). Hopanes were analysed with $M^+$ (412 for $C_{30}$ hopanes)$\rightarrow$ $m/z$ 191 and methylhopanes with $M^+\rightarrow m/z$ 205 multiple reaction monitoring transitions. 2-Methylhopane peak assignment was based on the retention time and the comparison with the Australian Geological Survey Organisation standard. Using a DB-5 MS capillary column, $C_{31}$ 2-methylhopane elutes just before $C_{30}$ $\alpha\beta$ hopane on the chromatogram.

**Pyrolysis.** Aliquots (1 mg) of diplopterol (courtesy of P. Adam and P. Schaeffer, University of Strasbourg), 5α(H)-cholestanol (≥95%, Sigma-Aldrich) or cholesterol (≥99%, Sigma-Aldrich) were transferred to glass tubes flame sealed at one end (Duran, 8 mm diameter, 1 mm wall thickness) dissolved in DCM. After DCM evaporation, ~20 mg active carbon (Sigma-Aldrich, DARCO, 100 mesh particle size, CAS: 7440-44-0) was added, tubes were evacuated to ≤300 mTorr and flame sealed with a gas torch. After pyrolysis in an oven at 300 °C for 24 h, tubes were cooled to room temperature, cracked open and the active carbon was transferred with sequential solvent rinses of $n$-hexane, DCM and methanol onto a small silica plug in a 4 ml solid phase extraction glass tube and subsequently extracted with about 10 ml $n$-hexane, 10 ml DCM and at least 4 ml methanol. Once solvents were evaporated under a stream of $N_2$, the pyrolysate was applied onto a silica gel microcolumn (about 500 mg in a glasswool-plugged Pasteur pipette), and the saturated hydrocarbon fraction was eluted with 1.5 dead volumes $n$-hexane followed by GC–MS analysis. Pyrolysates were analysed on a Thermo Quantum XLS Ultra triple-quadrupole MS coupled to a Thermo Trace GC Ultra fitted with a VF-1 MS column (40 m, 0.15 mm i.d., 0.15 μm film thickness).

### Reporting summary

Further information on research design is available in the Nature Portfolio Reporting Summary linked to this article.

## Data availability

All data needed to evaluate the conclusions in the paper are present in the paper and/or Supplementary Information. Additional and raw data and used sample material related to this paper may be requested from the authors.

## Code availability

The data and code used to execute cyanobacterial species tree constructions are available at https://github.com/David-Gold/2022_HpnP.

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

## Acknowledgements
We thank D. Edwards and the organic geochemistry team at Geoscience Australia for oil samples from the National Collection and the Geological Survey of Western Australia, the Northern Territory Geological Survey, T. Gallagher, E. Grosjean, K. Kirsimae, A. Lapland, L. Marynowski, M. Moczydłowska, S. Porter, N. Sheldon, E. Sperling and S. Zhang for rock specimens and extracts. We also thank P. Adam and P. Schaeffer (University of Strasbourg) for providing the diplopterol educt employed in the pyrolysis experiments and R. Tarozo (Max Planck Institute for Biogeochemistry) and J. Hope (The Australian National University) for assistance with GC–MS. This work was supported by DFG Priority programme 2237 and the Helmholtz Society to C.H.; Agouron geobiology postdoctoral fellowship to Y.H.; postdoctoral fellowship provided by the Central Research and Development Fund of the University of Bremen to B.J.N.; National Science Foundation grant 2044871 to D.A.G.; Australian Research Council grants DP160100607, DP170100556 and DP200100004 to J.J.B.; and National Institutes of Health grant R01AR069137, Human Frontier Science Program grant RGP0041, National Science Foundation grant 2032315 and Department of Defense grant MURI W911NF-16-1-0372 to E.A.G.

## Author contributions
Y.H. and B.J.N. conceived of the work. Y.H., E.A.G. and D.A.G. performed genetic data collection and analyses. B.J.N., G.V., L.M.v.M., C.B. C.H. and J.J.B. performed biomarker data collection and analyses. Y.H. and B.J.N. wrote the original draft. All authors participated in data interpretation, edited the draft and agreed to the published version of the paper.

## FundingInformation

## Competing interests
The authors declare no competing interests.

## Additional information
**Extended data** is available for this paper at https://doi.org/10.1038/s41559-023-02223-5.

**Correspondence and requests for materials** should be addressed to Yosuke Hoshino or Benjamin J. Nettersheim.

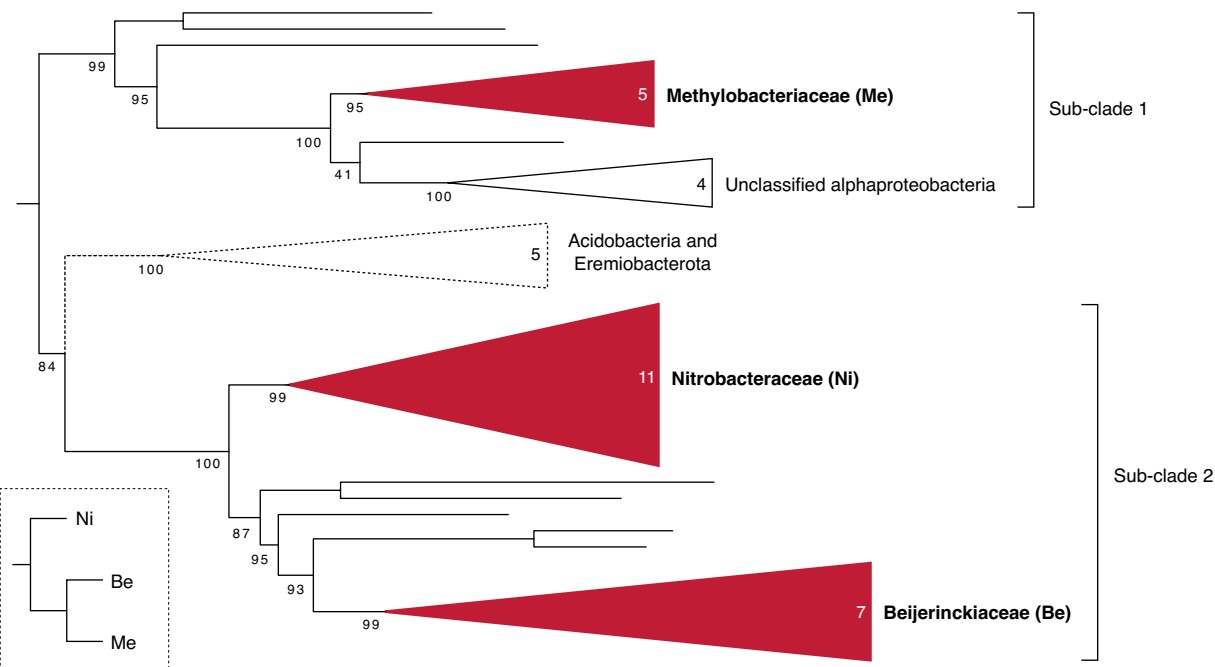

**Extended Data Fig. 1 | Evolutionary relationship of alphaproteobacterial HpnP proteins.** Three major HpnP-containing families are shown in the two alphaproteobacterial sub-clades within the HpnP tree (Methylobacteriaceae, Nitrobacteraceae and Beijerinckiaceae). The inset indicates the species relationship between the three Hyphomicrobiales families[32]. See Supplementary Fig. 1 for the complete tree with the species annotation.

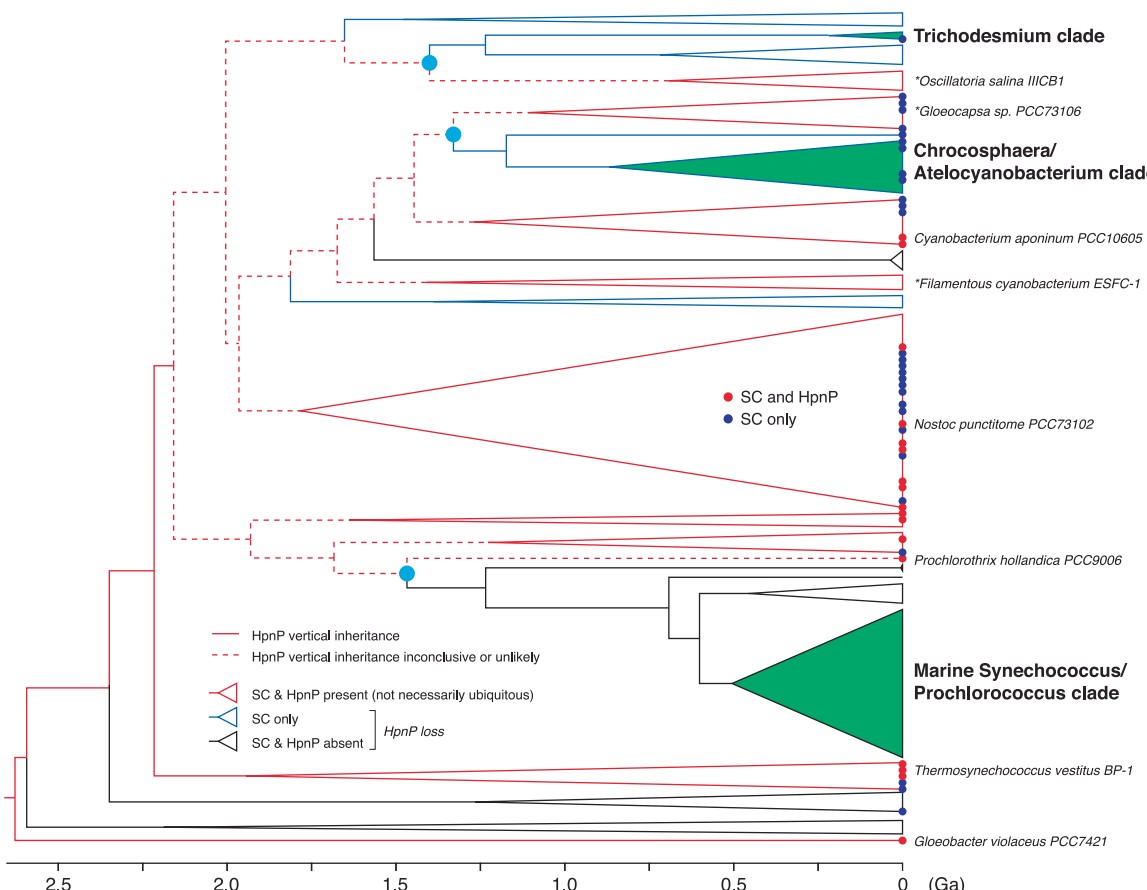

**Trichodesmium clade**

*Oscillatoria salina IIICB1*

*Gloeocapsa sp. PCC73106*

**Chrocosphaera/ Atelocyanobacterium clade**

*Cyanobacterium aponinum PCC10605*

*Filamentous cyanobacterium ESFC-1*

● SC and HpnP
● SC only

*Nostoc punctitome PCC73102*

*Prochlorothrix hollandica PCC9006*

— HpnP vertical inheritance
--- HpnP vertical inheritance inconclusive or unlikely

◁ SC & HpnP present (not necessarily ubiquitous)
◁ SC only
◁ SC & HpnP absent ⎤ *HpnP loss*

**Marine Synechococcus/ Prochlorococcus clade**

*Thermosynechococcus vestitus BP-1*

*Gloeobacter violaceus PCC7421*

2.5  2.0  1.5  1.0  0.5  0  (Ga)

**Extended Data Fig. 2 | Evolution of the extant lineages of marine planktonic cyanobacteria and its relationship to HpnP gene loss.** HpnP was likely lost long before the emergence of extant planktonic marine species. The dated species tree is adapted from a recent molecular clock study (see Supplementary Fig. 6 for the original figure with the species annotation)[54]. Presence/absence of HpnP in individual sub-clades was inferred from Supplementary Table 5 and recently-published phylogenomic studies of Cyanobacteria[55,56]. Inference for the vertical inheritance of the HpnP gene (solid line) is based on Fig. 3. Several representative HpnP-containing species are shown next to the corresponding clades. Asterisks indicate that the species is closely related to the corresponding clade, but is not included in the dataset in the cited study. Light blue circles indicate the possible timing of HpnP gene loss in the lineages that include modern marine planktonic species. It is noted that the timing of HpnP gene loss is possibly even earlier, depending on the actual HpnP evolutionary history in Cyanobacteria (dashed lines; vertical inheritance vs. horizontal transfer). Figure adapted from ref. 54 under a Creative Commons license CC BY 4.0.

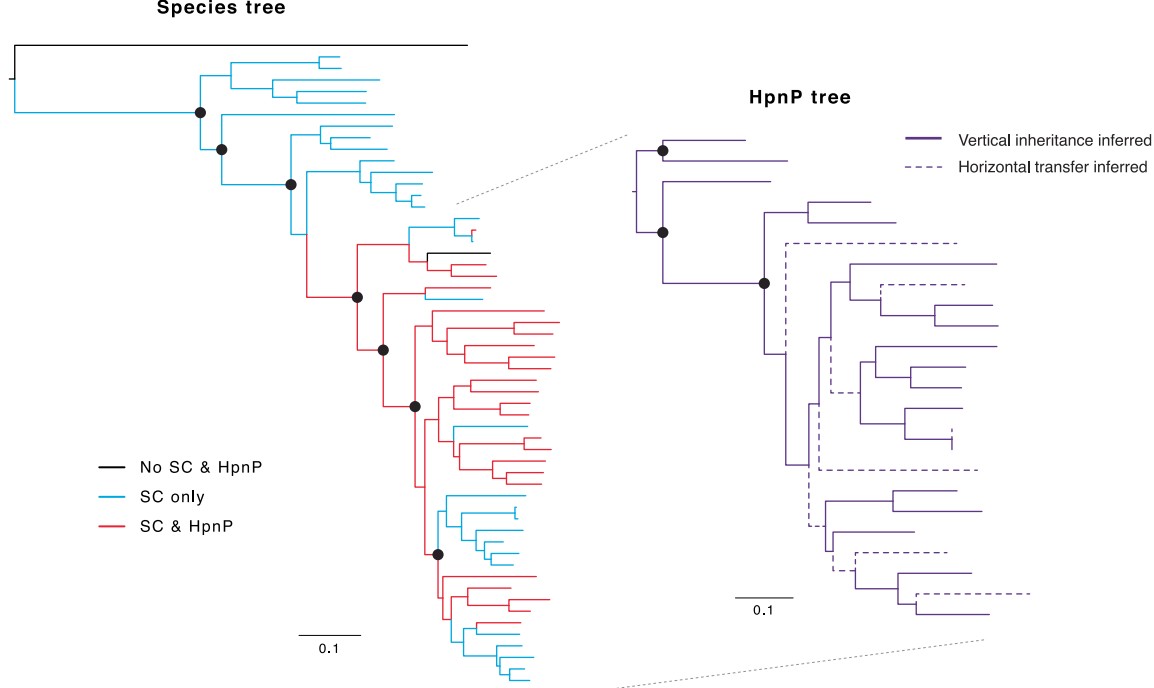

**Extended Data Fig. 3 | Distribution of HpnP in Rokubacteria.** A species tree that contained 56 representative rokubacterial species (Supplementary Fig. 7) was compared to the HpnP tree (Supplementary Fig. 8) through Notung reconciliation analyses (Supplementary Fig. 9). Solid lines in the HpnP tree indicate that they are consistent with the species tree. Dashed lines indicate that they are not consistent with the species tree and thus the presence of HGT is inferred by Notung. Filled black circles indicate that the node support is >85% (node support is shown only for major clades). The scale bar represents 0.1 amino acid replacements per site per unit evolutionary time.

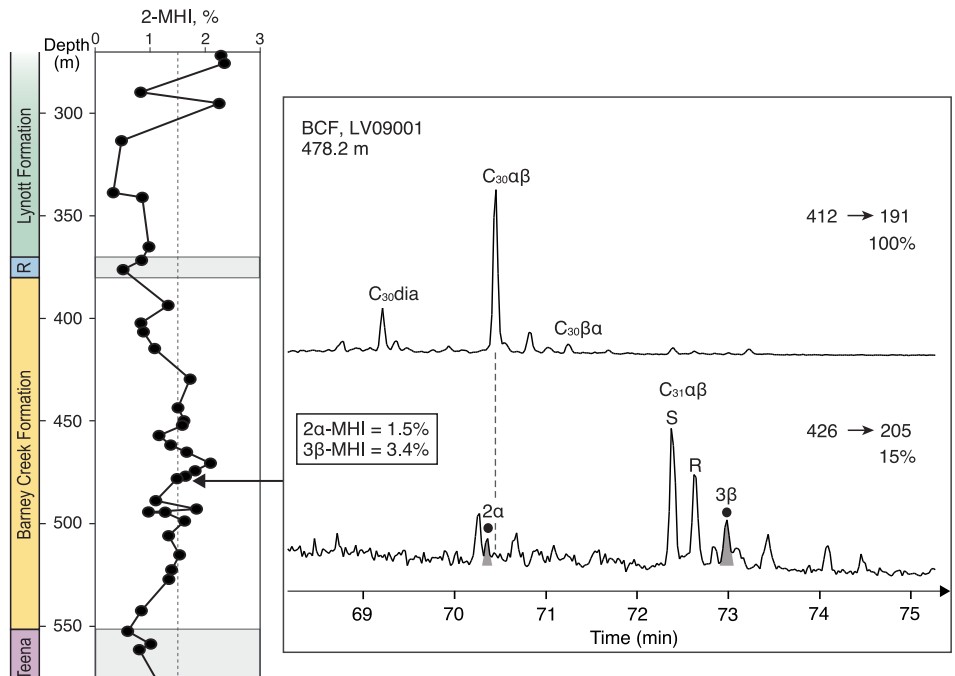

**Extended Data Fig. 4 | Stratigraphic trend of the 2-MHI.** 2-MHI values were calculated for the Paleoproterozoic LV09001 drill core from the McArthur Basin in Northern Australia ( ~ 1.64 Ga) (see also Supplementary Note 6). Representative chromatograms for non-methylated hopanes ($C_{30}$ diahopanes, $C_{30}$ & $C_{31}$ hopanes) and methylhopanes (2α and 3β) are shown (478.2 m depth). $C_{31}$ 2-methylhopane elutes just prior to $C_{30}$ αβ-hopane on a DB-5 MS capillary column. Chromatograms are annotated by Metastable Reaction Monitoring (MRM) precursor → products pairs. Percentages provide relative heights of the largest peaks in the chromatograms.

# Reporting Summary

## Statistics

For all statistical analyses, confirm that the following items are present in the figure legend, table legend, main text, or Methods section.

| n/a | Confirmed | |
|---|---|---|
| ☐ | ☒ | The exact sample size (*n*) for each experimental group/condition, given as a discrete number and unit of measurement |
| ☐ | ☒ | A statement on whether measurements were taken from distinct samples or whether the same sample was measured repeatedly |
| ☐ | ☒ | The statistical test(s) used AND whether they are one- or two-sided<br>*Only common tests should be described solely by name; describe more complex techniques in the Methods section.* |
| ☒ | ☐ | A description of all covariates tested |
| ☐ | ☒ | A description of any assumptions or corrections, such as tests of normality and adjustment for multiple comparisons |
| ☒ | ☐ | A full description of the statistical parameters including central tendency (e.g. means) or other basic estimates (e.g. regression coefficient) AND variation (e.g. standard deviation) or associated estimates of uncertainty (e.g. confidence intervals) |
| ☒ | ☐ | For null hypothesis testing, the test statistic (e.g. $F$, $t$, $r$) with confidence intervals, effect sizes, degrees of freedom and $P$ value noted<br>*Give P values as exact values whenever suitable.* |
| ☐ | ☒ | For Bayesian analysis, information on the choice of priors and Markov chain Monte Carlo settings |
| ☒ | ☐ | For hierarchical and complex designs, identification of the appropriate level for tests and full reporting of outcomes |
| ☒ | ☐ | Estimates of effect sizes (e.g. Cohen's *d*, Pearson's *r*), indicating how they were calculated |

*Our web collection on statistics for biologists contains articles on many of the points above.*

## Software and code

Policy information about availability of computer code

| Data collection | NCBI protein database, NCBI Identical Protein Group database. |
|---|---|
| Data analysis | Muscle v3.8.31, IQ-TREE v2.1.06, MrBayes v3.2.6, OrthoFinder v2.5.4, FASconCAT-G v1, NOTUNG v2.9. |

For manuscripts utilizing custom algorithms or software that are central to the research but not yet described in published literature, software must be made available to editors and reviewers. We strongly encourage code deposition in a community repository (e.g. GitHub). See the Nature Portfolio guidelines for submitting code & software for further information.

## Data

Policy information about availability of data

All manuscripts must include a data availability statement. This statement should provide the following information, where applicable:
- Accession codes, unique identifiers, or web links for publicly available datasets
- A description of any restrictions on data availability
- For clinical datasets or third party data, please ensure that the statement adheres to our policy

All data needed to evaluate the conclusions in the paper are present in the paper and/or the Supplementary Materials. Additional and raw data, as well as used sample material, related to this paper may be provided by the authors upon reasonable request.

# Human research participants

Policy information about studies involving human research participants and Sex and Gender in Research.

| | |
|---|---|
| Reporting on sex and gender | n/a |
| Population characteristics | n/a |
| Recruitment | n/a |
| Ethics oversight | n/a |

Note that full information on the approval of the study protocol must also be provided in the manuscript.

# Field-specific reporting

Please select the one below that is the best fit for your research. If you are not sure, read the appropriate sections before making your selection.

☐ Life sciences  ☐ Behavioural & social sciences  ☒ Ecological, evolutionary & environmental sciences

For a reference copy of the document with all sections, see nature.com/documents/nr-reporting-summary-flat.pdf

# Ecological, evolutionary & environmental sciences study design

All studies must disclose on these points even when the disclosure is negative.

| | |
|---|---|
| Study description | Trace organic analyses of geological samples |
| Research sample | Sedimentary rocks from drill cores and outcrops. |
| Sampling strategy | To obtain a geological trend of 2-methylhopane abundances across the Proterozoic, we exclusively (re)analysed drill core and rock samples already available in the sample collection of the Hallmann and Brocks labs that were known from previous analyses to host indigenous biomarkers. |
| Data collection | B.J.N., G.V., L.v.M. C.B., and J.J.B. collected and recorded the data. |
| Timing and spatial scale | We generally tried to include representative samples for each formation by sampling at different stratigraphic heights (depending on formation thickness meters to tens of meters apart) and including different lithologies (e.g. carbonates and shales). Rock samples typically comprise 1 to 3 cm sedimentary thickness and thus typically integrate tens to thousands of years of depositional history depending on local deposition rates. |
| Data exclusions | Samples that contain anthropogenic contamination are excluded as described in the Supplementary Information. |
| Reproducibility | Multiple data points are obtained from individual formations. |
| Randomization | n/a |
| Blinding | n/a |

Did the study involve field work?  ☒ Yes  ☐ No

# Field work, collection and transport

| | |
|---|---|
| Field conditions | Field conditions were not meaningful for the collection of rock samples. |
| Location | Multiple locations from Eurasia, Australia, Africa and America. The details are in Supplementary Information. |
| Access & import/export | Required sample permits were obtained in advance for all outcrop and drill core store sampling campaigns previously conducted by the Brocks and Hallmann groups. Samples were shipped to the laboratory according to local costums regulations. |
| Disturbance | Most samples where specifically kept in (publically funded) drill core stores for the purpose scientific sampling and analyses. Only |

| Disturbance | subsamples of core material were taken to minimize disturbance of drill cores. Outcrops samples were obtained by hand (hammer) only with minimal disturbance to the environment. |
| --- | --- |

# Reporting for specific materials, systems and methods

We require information from authors about some types of materials, experimental systems and methods used in many studies. Here, indicate whether each material, system or method listed is relevant to your study. If you are not sure if a list item applies to your research, read the appropriate section before selecting a response.

## Materials & experimental systems

| n/a | Involved in the study |
| --- | --- |
| ☒ ☐ | Antibodies |
| ☒ ☐ | Eukaryotic cell lines |
| ☒ ☐ | Palaeontology and archaeology |
| ☒ ☐ | Animals and other organisms |
| ☒ ☐ | Clinical data |
| ☒ ☐ | Dual use research of concern |

## Methods

| n/a | Involved in the study |
| --- | --- |
| ☒ ☐ | ChIP-seq |
| ☒ ☐ | Flow cytometry |
| ☒ ☐ | MRI-based neuroimaging |

