## [Peer Review File · Nature Ecology & Evolution]

Peer Review Information

Journal: Nature Ecology & Evolution

Manuscript Title: Genetics re-establish the utility of 2-methylhopanes as cyanobacterial biomarkers before 750 million years ago

Corresponding author name(s): Yosuke Hoshino, Benjamin J. Nettersheim

Editorial Notes:

Reviewer Comments & Decisions:

Decision Letter, initial version:

14th March 2022

Dear Dr Hoshino,

Thank you very much for your enquiry about submitting your manuscript "2-Methylhopanes as cyanobacterial specific biomarkers before 900 million years ago" to Nature Ecology & Evolution. It certainly sounds interesting, and we would be happy to consider it for publication. However, I'm sure you'll understand that we cannot make a firm decision about whether to send the paper out to review until we have carefully read the full paper (and appropriate background literature).

In order to submit your complete manuscript to Nature Ecology & Evolution, please use the link below:

[REDACTED]

If you have any questions, please feel free to contact me.

Yours sincerely,

[REDACTED]

Decision Letter, first revision:

3rd January 2023

Dear Dr Hoshino,

Your manuscript entitled "Genetics re-establish the utility of 2-methylhopanes as cyanobacterial biomarkers before 750 million years ago" has now been seen by four reviewers, whose comments are attached. The reviewers have raised a number of concerns which will need to be addressed before we can offer publication in Nature Ecology & Evolution. We will therefore need to see your responses to the criticisms raised and to some editorial concerns, along with a revised manuscript, before we can reach a final decision regarding publication.

As you will see, the reviewers are largely positive about the manuscript but do have a number of

2technical and structural comments on how it might be improved. In particular, reviewer 2 reveals that they have reviewed this manuscript previously elsewhere, and while some aspects of the paper have improved, they remain concerned about the focus on the marine sedimentary record of 2-Methylhopanes, as well as the discussion of previous work--on the latter point, while we agree editorially with the reviewer that it's important to contextualise the work in light of previous research, we are concerned that a blow-by-blow account of previous research on 2-Methylhopanes (rather than a summarised account) might add rather too much length to the manuscript. To that end, when you are revising in response to reviewer 2's comments, while we do need you to address all of them, you might consider whether some of the requested discussion is more appropriate to the Supplementary Information, leaving a more summarised account in the main text. In addition to the other general comments of reviewers, please also pay particular attention to reviewer 3's technical comments on phylogenetics.

We therefore invite you to revise your manuscript taking into account all reviewer and editor comments. Please highlight all changes in the manuscript text file.

- * Include a "Response to reviewers" document detailing, point-by-point, how you addressed each reviewer comment. If no action was taken to address a point, you must provide a compelling argument. This response will be sent back to the reviewers along with the revised manuscript.
- * If you have not done so already please begin to revise your manuscript so that it conforms to our Article format instructions at <http://www.nature.com/natecolevol/info/final-submission>. Refer also to any guidelines provided in this letter.
- * Include a revised version of any required reporting checklist. It will be available to referees (and, potentially, statisticians) to aid in their evaluation if the manuscript goes back for peer review. A revised checklist is essential for re-review of the paper.

[REDACTED]

2Nature Ecology & Evolution is committed to improving transparency in authorship. As part of our efforts in this direction, we are now requesting that all authors identified as 'corresponding author' on published papers create and link their Open Researcher and Contributor Identifier (ORCID) with their account on the Manuscript Tracking System (MTS), prior to acceptance. ORCID helps the scientific community achieve unambiguous attribution of all scholarly contributions. You can create and link your ORCID from the home page of the MTS by clicking on 'Modify my Springer Nature account'. For more information please visit www.springernature.com/orcid.

[REDACTED]

Reviewer expertise:

Reviewer #1: pre-Cambrian rock record

Reviewer #2: organic biogeochemistry

Reviewer #3: phylogenetics and comparative genomics

Reviewer #4: organic biogeochemistry

Reviewers' comments:

Reviewer #1 (Remarks to the Author):

2-Methylhopanes played a key role in discussions about the long-term evolution of marine ecology for years. Over the past few years there has been increasing concern and discussion about what these biomarker records can tell us. In my opinion this is an exciting paper that puts these biomarkers back on 'center stage' in efforts to make sense of Precambrian oceans. The paper is well-written and the results seem robust. I have no doubt that this paper will spark additional work – but I think this is exactly the sort of exciting work on Earth's history that NEE should be targeting. With that very positive sentiment expressed, I do have a few comments, questions and concerns.

1) The uptick in the 2-methylhopanes in the fossil record is linked to the HpnP-containing Alphaproteobacteria (Hyphomicrobiales). However, I am not seeing a reason why this couldn't alternatively be linked to a shift to more productive (more P rich) and more oxic oceans. The authors

3stress that 2-methylhopanes can be concentrated when there is a extensive aerobic degradation—that is exactly what we would expect moving into the Neoproterozoic and Phanerozoic.

2) I would like to see more information about the marine ecology of hyphomicrobiales. For instance what portion of the microbial biomass are HpnP-containing Alphaproteobacteria in typical marine environments? Based on this information is reasonable to suggest they would be a the dominate source of 2-methylhopanes?

3) The updated biomarker record makes sense to me – and including it makes the paper much stronger. But it is entering into a fairly debated topic with very limited discussion in the main text, which I do not think will help move forward debate. I think this section needs to be further flushed out. For instance, I would say there needs to be a clear statement about the presence of robust, verifiable 'typical' steranes in the Precambrian record.

4) I like the B12 hypothesis, this section is a strength of the paper. But I was left looking for more information. I would suggest taking text to ref

5) This is a small point but I am not sure I would use symbiosis when talking about algae and B12 producing bacteria. There is some variability in how the term is used – but there are some strict definitions.

6) It would be good to see more information about method for the pyrolysis experiments

Reviewer #2 (Remarks to the Author):

The manuscript by Hoshino & Nettersheim et al. describes extensive efforts to 1) produce an uncontaminated Proterozoic record of 2-methylhopanes (putative markers for cyanobacteria in the rock record) and 2) to re-evaluate the specificity of 2-methylhopanes as cyanobacterial biomarkers using phylogenetic analyses. I had previously reviewed an earlier version of this manuscript for a different journal. Some of the issues raised earlier still persist in the present submission. The authors will thus note some overlap between the two reviews. In the present manuscript, the authors conclude that 2-methylhopanes were present at low levels during the Proterozoic and that before ~750 Ma the most likely source of hopanoids would have been cyanobacteria. The main evidence presented is the inferred vertical inheritance of hpnP in Cyanobacteria including early branching lineages, which contrasts with horizontal inheritance in certain lineages of alphaproteobacteria and thus places the first occurrence of hpnP in cyanobacteria much earlier than in alphaproteobacteria. This work is significant as it aims to resolve the controversy whether 2-methylhopane biosynthesis (via the hpnP gene) originated in cyanobacteria or alphaproteobacteria and which of these groups predominantly contributed to the 2-methylhopane signal in the rock record. The authors make a case for a predominantly cyanobacterial source prior to 750 Ma and a primarily alphaproteobacterial source after 750 Ma.

I support publication of this work. However, I find two major issues that should be addressed:

1) The authors focus on the marine sedimentary record of 2-methylhopanes. The problem that 2-methylhopanes are not actually found in extant marine cyanobacteria (Elling et al., 2020; Naafs et al.,

42021) needs to be discussed in detail (also in the light of evolution of marine cyanobacteria) as this suggests that the 2-methylhopane geologic record is dependent on depositional setting (e.g., near-shore vs. pelagic) in addition to geologic time (timing of cyanobacterial radiations). I think it is important to discuss this more prominently as this will be a very important aspect for future studies. This discussion is also warranted because the authors exclude Rokubacteria as a source of 2-methylhopanes in marine environments because they have no marine representatives but the same is actually true for hpnP-containing Cyanobacteria. This discussion is also very important for interpreting the assumed shift in 2-methylhopane sources inferred to have occurred around 750-500 Ma. This timeframe was significant for the evolution of cyanobacteria as it coincides with the emergence of extant marine cyanobacterial lineages (Sánchez-Baracaldo, 2015), all of which do not contain 2-methylhopanes. Why was 2-methylhopane biosynthesis lost during that radiation if it was previously a pervasive feature? Importantly, since coastal environments may receive 2-methylhopanoid input from freshwater or tidal environments, I wonder if the shift in 2-methylhopane abundance during that time is biased by different depositional settings for these records?

2) In my view, the authors do not adequately discuss prior work. The manuscript builds upon the long-standing controversy surrounding the cyanobacterial vs. alphaproteobacterial origin of 2-methylhopanes. Therefore, the arguments and merits of previous studies should be discussed and refuted individually, not summarily as presently done (particularly Ricci et al., 2015). The authors seem to imply that the majority of cyanobacterial lineages/species/genomes contain SC and hpnP, whereas previous studies have found the opposite to be true (Ricci et al., 2015; Elling et al., 2020; Naafs et al., 2021), see also (see also Fig. 6 in Kusch and Rush, 2022). Particularly, these earlier studies found that hpnP is not present in marine cyanobacteria, even in environments that favored freshwater/brackish lineages of cyanobacteria. Finally, the phylogeny of hpnP derived by the authors differs to some extent from previous studies, some of which placed the cyanobacterial hpnP sequences within alphaproteobacteria. How would these alternative phylogenies impact the conclusions? A more robust discussion on the choice of substitution model/phylogenetic methods and comparison to previous works is warranted here. The lack of this discussion is surprising, given that this formed a major part of a previous version of this manuscript.

Line comments:

Line 94-96: Occurrence in Actinobacteria described in Naafs et al. 2021 (Fig. 5 and supplement) and Elling et al. (2022, Fig. S2)

Line 107-110: The physiological roles of hopanoids have been explored in some detail (e.g., Doughty et al., 2009; Wu et al., 2015).

Line 204-207: This argument was previously made by Ricci et al. (2015), based on earlier work by Summons, which should be credited here.

Line 270-271: Here or elsewhere the depositional setting of the McArthur basin formation should be discussed in the light of possible biases in the 2Me record, since hpnP-containing cyanobacteria are mostly freshwater or benthic/coastal species.

5Line 346-350: This statement is speculative and could be toned down.

References

- Doughty, D.M., Hunter, R.C., Summons, R.E., Newman, D.K., 2009. 2-Methylhopanoids are maximally produced in akinetes of *Nostoc punctiforme*: geobiological implications. *Geobiology* 7, 524–532.
- Elling, F.J., Hemingway, J.D., Evans, T.W., Kharbush, J.J., Spieck, E., Summons, R.E., Pearson, A., 2020. Vitamin B12-dependent biosynthesis ties amplified 2-methylhopanoid production during oceanic anoxic events to nitrification. *Proceedings of the National Academy of Sciences* 117, 32996–33004.
- Kusch, S., Rush, D., 2022. Revisiting the precursors of the most abundant natural products on Earth: A look back at 30+ years of bacteriohopanepolyol (BHP) research and ahead to new frontiers. *Organic Geochemistry* 172, 104469.
- Naafs, B.D.A., Bianchini, G., Monteiro, F.M., Sánchez-Baracaldo, P., 2021. The occurrence of 2-methylhopanoids in modern bacteria and the geological record. *Geobiology* n/a. doi:10/gk9j7d
- Ricci, J.N., Michel, A.J., Newman, D.K., 2015. Phylogenetic analysis of HpnP reveals the origin of 2-methylhopanoid production in Alphaproteobacteria. *Geobiology* 13, 267–277.
- Sánchez-Baracaldo, P., 2015. Origin of marine planktonic cyanobacteria. *Scientific Reports* 5, 17418.
- Wu, C.-H., Bialecka-Fornal, M., Newman, D.K., 2015. Methylation at the C-2 position of hopanoids increases rigidity in native bacterial membranes. *eLife* 4, e05663.

Reviewer #3 (Remarks to the Author):

Hoshino et al. study the evolution of the genes underpinning 2-methylhopane biosynthesis in Bacteria. Their analyses aim to address the question of whether these molecules can be used as biomarkers for Cyanobacteria, given the presence of the biosynthetic genes in some other bacterial phyla (Alphaproteobacteria and Acidobacteria). The analyses suggest that these genes were acquired recently by HGT in Alphaproteobacteria, suggesting that the presence of the lipids earlier in the rock record likely reflects the presence of Cyanobacteria --- rehabilitating 2-methylhopanes as biomarkers for early cyanobacterial evolution. The authors then go on to re-appraise the biomarker record for these lipids, including new analyses of the interior and exterior of samples to distinguish original presence from contamination from more recent rocks.

Overall, this is an interesting manuscript, and the combination of phylogenetic analyses and geobiological/biomarker experiments is welcome. My detailed remarks are restricted to the phylogenetic analyses, which I am (arguably) competent to assess, but the overall impression is that this paper reports an integrated body of work that makes an important contribution to our understanding of the early bacterial (molecular) fossil record. However, there seem to be some important gaps in the analyses that could be plugged fairly straightforwardly, and would help to shore up the authors' conclusions, as I detail below.

Major comments

6- I appreciate that the authors used gene tree-species tree reconciliation to study the evolution of the key HpnP gene in Cyanobacteria. However, I have a couple of questions about these analyses. First, why was the gene tree reconciled against a species tree that only contained Cyanobacteria that have HpnP? It seems like this analysis will greatly under-estimate the number of losses. For example, the evidence for the gene being present ancestrally in Cyanobacteria would be stronger if the gene is broadly, rather than sporadically, distributed in the clade. On a related note, how does Notung determine the probability (or, I suppose, support) for a gene to be present at the root of the species tree --- for example, how much less parsimonious were scenarios in which the gene was acquired later within Cyanobacteria and then passed around by transfer? Bottoming out these issues is important because - as I understand it - it is the fit between the gene and species tree that the authors are using to date the gene acquisition (by dating the root of the Cyanobacteria in the species tree).

- On a related note, why not perform the same analysis for Alphaproteobacteria? Presumably the results will be quite different to the cyanobacterial case (i.e. the reconciliation method will map the gene to a much shallower node on the dated species tree), providing an empirical test of the hypothesis that the gene traces deeper in Cyanobacteria than in Alphaproteobacteria.

- The HpnP gene tree suggests that the gene may also be rather ancient in Rokubacteria (at least, there is a substantial clade of homologues in that phylum). How does the gene tree here compare to the species tree? The authors mention that they think it unlikely that these bacteria had much of a role in generating the lipids early in evolution, but do not provide a great deal of evidence for this. It would be worthwhile performing the same analysis here as for Cyanobacteria (and, in my view, Alphaproteobacteria) --- is the gene transferred a lot in Rokubacteria, or does it date to their common ancestor? I appreciate that the authors have done quite a lot of work, and may not wish to (for example) estimate the age of the Rokubacterial ancestor using molecular clocks, but at least testing whether the gene might have been present at their root or not seems quite important to the main question being tested here. For example, if the gene actually seems to be recently acquired (as in Alphaproteobacteria), the authors' case that only Cyanobacteria are likely to have contributed to the biomarker record early on would be greatly strengthened. As it stands, the question of Rokubacteria seems to be something of a loose thread.

Minor comments

- "transferred vertically" - avoid, maybe "inherited vertically".
- line 156: Eremiobacterota misspelled?
- I would modify the Figure 4 legend slightly to indicate that both the topology and the inferred species tree dates are based on the recent study cited.

Reviewer #4 (Remarks to the Author):

General comments,

This is a nice work presented by Hoshino et al., which puts a genetic constrain on the interpretation of fossil 2Me-hopanoids in geological record. The BLAST search of HpnP gene in additional genomes,

7phylogenetic analysis of identified homologs, and the molecular clock estimates all yield valuable information for understanding the evolution history of 2Me-hopanoids producing microbes. And, the revised 2-MHI dataset that eliminates bias of contamination is a great contribution to the field.

My only critique lies with the authors' interpretation on the high 2-MHI of Phanerozoic. Naafs et al., (2021, *Geobiology*) conducted very similar works, including phylogenetic analyses of HpnP, biological production of 2Me-hopanoids and the geological record of 2-MHI and related the Phanerozoic 2Me-hopanoids distribution to marine nitrogen cycle. I don't know why Naafs' work is not mentioned in this study. It is nice that Hoshino has proposed a potential connection between the rise of algae and B12 producing alphaproteobacteria in the Phanerozoic, but the coincidence of extreme high 2-MHI with marine anoxic events and the ecological role of alphaproteobacterial in marine nitrogen cycle can be further discussed. Another recent publication, Garby et al. (2022, *EMR*), reported the dominant 2Me-hopanoids contribution by cyanobacteria over alphaproteobacteria under harsh/extreme conditions. Additional discussion on the role of environmental stress influencing the geological distribution of 2Me-hopanoids can also enrich this paper.

Naafs, B.D.A., Bianchini, G., Monteiro, F.M., Sánchez-Baracaldo, P., 2021. The occurrence of 2-methylhopanoids in modern bacteria and the geological record. *Geobiology* 20, 41–59.
Garby, T. J., Jordan, M., Timms, V., Walter, M. R., & Neilan, B. A. (2022). 2-Methylhopanoids in geographically distinct, arid biological soil crusts are primarily cyanobacterial in origin. *Environmental Microbiology Reports*, 14(1), 164-169.

Specific comments,

Line 39, typo, change 'form' to 'from'

Line 93-94, only 27 phyla containing SC and 10 of HpnP are shown in Figure 1

Line 104-105, it is not clear to me how this is inferred.

Line 131-133, Any studies have reported or discussed fossil hopanoids recycling by microbe?

Line 181, please provide the estimated percentage of gene loss.

Line 207-214, these two sentences do not fit very well to the context of this subsection. These are discussions of general biomarker preservation and detection, nothing specific of 2Me-Hopane. GC-APLI-TOF was only reported to be highly sensitive for polyaromatic hydrocarbons detection, probably because they are preferentially ionized than others.

Line 230, delete "and possibly Rokubacteria", because there is no data to support this. The following discussion on their unlikely contribution to fossil 2-Me hopane is good enough.

Line 247, please refer to Fig. 4A in the bracket.

The calculation of these node bars (a, b and c) is very important for this work. However, the nodes estimation is neither well described in text nor the caption of Fig. 4. It is referred to Fig. S6, but no such information provided in Fig. S6.

Line 253, those triangles are in red not brown in Fig. 4A.

Line 636, do nodes a, b and c represent Be, Ni and Me, respectively?

Last sentence of Supplementary Text 3

8“, which do not possess a functional group in the A-ring,”

*****END*****

Author Rebuttal, first revision:Reviewer #1 (Remarks to the Author):

The paper is interesting as it shows that a gene, by simply generating a dorso-ventral gradient early in development, is somehow involved in hair placode formation but does not provide an explanation for stripe formation. The same gene, by differential expression in dark and light stripes later, participates in the color differences. That patterning and melanocyte pigment generation can be mediated by the same gene is not without precedence and thus is believable. Although the work is significant and the model presents compelling interest for developmental and pigmentation biology, I would recommend against publication in its current form for the following reasons:

1. The study involves a considerable amount of experiments and generated data however the reader is directly funneled to a single gene in Figure 1.

We perform a completely unbiased RNAseq experiment and data analyses to find factors correlated with the unusual pattern of hair placode formation seen in striped mice, compared to lab mice. After we perform our data analyses, we explicitly state our rationale for focusing on *Sfrp2* (page 3). We then perform a large variety of downstream experiments aimed at uncovering the function of this gene in the context of placode formation and pigmentation. In the discussion, we state there are additional players and that *Sfrp2* is not the only factor involved in regulating the phenotype we are studying (page 7).

We decide to focus on a single gene because this allows us to pursue downstream mechanistic experiments aimed at understanding our phenotype. This makes the problem at hand more tractable, as all these experiments would be difficult to carry with multiple genes.

It is worth pointing out that most studies examining the genetic/developmental basis of a trait follow similar strategies and, like ours, the implication is never that there is only one gene involved in regulating a specific phenotype. To list but a few examples: Stickleback studies involving *Eda* (armor plates) or *Pitx1* (pelvic spines) or *Hoxd11b* (spines); Pigmentation studies involving the gene *Mc1r* (mice) or *Agouti* (mice) or *Dkk4* (cats); Beak development studies in Darwin's finches involving the gene *Alx1*; Loss of limbs in

snakes involving the gene *Shh*; Butterfly wing patterning involving the gene *Optix* or *WntA*, etc.

2. The gene expression data provided in the supplementary table show the same “baseMean” regardless of the samples considered in the individual pairwise comparison.

Yes, because baseMean values are calculated as the mean expression of a gene across all tissues being investigated. For this reason, this value is always the same for a given gene, regardless of which two regions are being compared.

3. What should be unbiased data analysis in experiments related to Figure 2 directly focuses on the effect of that single gene.

This is incorrect. The bulk RNA-seq screen and data analyses were done in an unbiased way. The rationale for focusing on *Sfrp2* is explained in our response above. Fig 2 describes experiments that were done *after* selecting our top candidate.

4. Ablation of *Sfrp2* subsequently does not show the phenotypic outcome theorized in Figure 3 when depleted in vivo: the stripe pattern is not prevented. The model should hence be adapted according to the authors' observations.

This observation is incorrect. The modelling does not predict that the stripe pattern will be prevented, as the reviewer mentions. Our model predicts subtle changes in stripe width in response to perturbation of the gradient, matching the phenotypes we observe. This is explicitly stated in Figure 3c and in the text (Page 5):

We next used additional simulations to study the relationship between alterations to the modulator gradient and potential consequences on stripe spacing. Our results demonstrated the continued stability of stripes, regardless of the magnitude of the modulator gradient (Extended Data Fig. 4b). More strikingly, however, we found that altering the strength of the modulator gradient led to stereotyped changes in the spacing of the stripes: in all models, changing only the magnitude of the modulator gradient produced subtle changes in stripe width (Fig. 3c, d and Extended Data Fig. 4b).

5. The authors focus their attention on the developing hair placodes in the first part of the study but fail to depict any related embryonic features upon *Sfrp2* KO later in the manuscript. Further, part of the coat phenotype is described as the differential hair length between the dark and light stripes (likely due to the differences in temporal onset of hair follicle formation). No mention of this feature is looked at upon *Sfrp2* KO.

The model we present suggests that *Sfrp2* narrows/expands the regions of placode formation that foreshadow the eventual striped pattern. The model does not argue that *Sfrp2* changes the timing between when hair placodes emerge in certain regions and emerge later in others. For this reason, the differential hair length between stripes is not expected to change in response to *Sfrp2* KO. The hair length phenotypes were presented to describe the phenotype and show that there is differential placode formation in the dorsal skin, but this is something not expected to change with perturbation of *Sfrp2*.

Based on the way we present our data, we acknowledge that this is something a reader may not infer easily (as evidenced by a related comment from reviewer 2). Thus, we will adjust the text to clarify this point.

With regards to embryonic phenotypes, this criticism is fair and we have ways of addressing it.

6. The link between the embryonic features depicted in Figure 1 between embryonic stripes R1, R2, R3 and R4 is lost when the authors change their nomenclature in Figure 4, acknowledging that embryonic R1 is, in fact, further divided in Midline (light stripe) and the actual R1 dark stripe, both of which lack embryonic placodes at the times they focused their interest.

This observation is correct and valid. However, the midline region and black stripe are indistinguishable at embryonic stages (they are adjacent, placode-less regions without pigment). In postnatal stages, they can be clearly distinguished. We perform measurements on both regions because both show quantifiable changes in width in response to *Sfrp2* knockout. Changes in width among regions is predicted by our mathematical model.

7. The stripes of the *Rhabdomys pumilio* are located on the trunk skin, but not in the head and neck region. These intriguing differences intrinsic to the model organism were not investigated or addressed.

In striped mouse, stripes are only found in the dorsal/trunk skin and our goal was to investigate how this phenotype is generated. Previous studies in laboratory mice have shown that the dorsal skin domain is developmentally and functionally distinct from the cranial skin domain. Furthermore, cranial hairs are normally shorter than dorsal hairs and the hair cycle in cranial hair is shorter than in dorsal hair, indicating that the biology of cranial hair follicles is different from that of dorsal follicles: By extension, it is likely that other aspects of cranial hairs, including pigmentation, are different too.

The differences between cranial and trunk domains are well established and have been studied in different contexts (e.g., <https://elifesciences.org/articles/22772>), <https://pubmed.ncbi.nlm.nih.gov/18713754/>, <https://pubmed.ncbi.nlm.nih.gov/30122476/>).

In addition, our own data shows that the biology of cranial and dorsal hair follicles is fundamentally different: at E16.5, the stage where we observe differences in hair follicles in dorsal skin, there are no hair follicles in the cranial skin region (see **Figure 2**).

For all these reasons, while it is interesting that there are no stripes on the head region, we don't think this is unexpected or surprising. We will make this point clear in a new version of the manuscript.

Figure 1. At E16.5, there are no hair follicles in the cranial region, while those in the dorsal skin are evident.

8. The authors use an unregulated ubiquitous overexpression of *Sfrp2* in laboratory mice, in an attempt to phenocopy their striped animal model, while they argue that *Sfrp2* is specifically expressed in skin fibroblasts, and that it is expressed in a gradient. Wnt signaling suppression is already known to repress hair follicle placode formation.

This comment reflects an incorrect understanding of this experiment and the data presented.

First, we did not drive 'unregulated ubiquitous' expression of *Sfrp2* in laboratory mouse, as the reviewer suggests. Instead, we used a dermal fibroblast specific promoter (i.e., Dermo-Cre) to drive expression only in '*dermal fibroblasts*', as these are the cells that express *Sfrp2* in striped mouse.

Second, this experiment was not designed to 'phenocopy' the striped mouse model, as the reviewer implies. It was designed *exclusively* to test whether expression of *Sfrp2* in dermal fibroblasts would inhibit hair follicle formation, which hasn't been investigated previously. At no point did we intend to mimic the dorsoventral expression gradient seen in striped mice. This would be very interesting to do, but it would be technically very challenging and is an experiment that addresses a different question.

The reviewer is correct that Wnt signaling suppression is already known to regulate placode formation. However, the role of *Sfrp2* in skin development has not been investigated. Since *Sfrp2* can lead to the activation or inhibition of Wnt signaling depending on cellular microenvironment (van Loon et al 2021), we felt it was important to test its mechanism of action in rodent skin and verify that, indeed, it was inhibiting Wnt signaling. Thus, the goal of this experiment was to test whether *Sfrp2* was acting as a Wnt inhibitor, not to test whether inhibiting Wnt signaling would lead to a reduction in hair follicle formation. This is explicitly stated in Page 4.

9. Further, the data representation is several instances inadequate, such as:

a. Bulk RNAseq: No FDR or FC cutoff information is provided to define differentially expressed genes, besides the pValue criteria. No detailed depiction of the data is provided, even as supplementary material.

This is incorrect. In Supplementary Table 1 we present '*adjusted P value*', which is the same as FDR. In a previous critique, this reviewer references this same supplementary table in reference to "baseMean" values, so we are surprised this wasn't evident.

b. Representative histology of the various dissected R stripes would be useful in supplementary data.

We would be happy to include these data.

c. scRNAseq: Anatomically distinct parts of the embryo were down sampled and pooled during for sequencing. Why? This experiment would allow to distinguish how various cell types such as keratinocytes, fibroblasts or melanocytes are

affected based on their anatomical location. This information is lost upon pooling them.

We chose to pool the data to obtain an unbiased characterization of striped mouse dorsal skin. We acknowledge the reviewer's point that not pooling the data provides information that can be used to establish differences occurring between the stripes.

In the new version of the manuscript, we will use stripe-specific single cell data to (1) examine markers of hair follicle formation and confirm there are differences in placode markers between the different stripes; and (2) to perform stripe-specific quantification of dermal fibroblast *Sfrp2* expression. This will allow us to show the average expression levels of *Sfrp2* in each stripe as well as the number of *Sfrp2* expressing fibroblasts in each stripe (please see our response to reviewer 2 on a related request).

d. The color palettes of the tSNE plot and its legend don't match and make the annotation irrelevant.

We do not understand this comment. From what is shown in Fig 2a, the colors and annotations *do* match.

e. If only the fibroblast population is of interest in this experiment, unsupervised subclustering and analysis of the fibroblast population should be performed.

This is exactly what we did. We performed unsupervised subclustering and analysis of fibroblasts. The graphs shown in Figs 2j, k, are generated after subclustering and analysis of fibroblasts is performed. We thought this was clear but would be happy to state this point more explicitly in the text.

f. Sample allocation information should be given in supplementary.

We are not entirely sure what the reviewer refers to with 'sample allocation'. In the methods, we explain how samples were obtained, processed, and we provide the number of embryos used per time point. In addition, figure legends contain sample numbers.

g. Macroscopy: no scale bar is provided for all the macroscopic pictures

We would be very happy to add these.

h. Microscopy: magnification is inappropriate to depict cellular resolution of gene expression.

We don't understand what is being referenced here. The insets shown in Fig 2F, H show cellular resolution. Moreover, all our imaging data is complemented with single cell data, which provides cellular resolution.

i. Gene legend doesn't match *Ctnnb1* expression images.

We apologize for this mistake and will be happy to fix it.

Reviewer #1 (Remarks to the Author):

2-Methylhopanes played a key role in discussions about the long-term evolution of marine ecology for years. Over the past few years there has been increasing concern and discussion about what these biomarker records can tell us. In my opinion this is an exciting paper that puts these biomarkers back on ‘center stage’ in efforts to make sense of Precambrian oceans. The paper is well-written and the results seem robust. I have no doubt that this paper will spark additional work – but I think this is exactly the sort of exciting work on Earth’s history that NEE should be targeting.

1

Helmholtz Centre Potsdam · GFZ German Research Centre for Geosciences · Public Law Foundation
Telegrafenberg · 14473 Potsdam · www.gfz-potsdam.de
Board of Trustees: MinDirig’in Oda Keppler (Chair)
Executive Board: Prof. Dr. Niels Hovius (Spokesman, interim) · Dr. Stefan Schwartz
Payments: Account No. 3093887 · Deutsche Bank · BLZ 120 700 00
IBAN DE8612070000309388700 · BIC Code DEUTDE33HAN · Tax No. 046/149/01166 · VAT DE138407750

HELMHOLTZ

15RESPONSE: We appreciate the reviewer's positive comment and the judgement that it is particularly suited for publication in *Nature Ecology & Evolution*. We hope that this study may become the fundament for time-resolved biomarker analyses, which will be an important future direction of biomarker research.

With that very positive sentiment expressed, I do have a few comments, questions and concerns.

1) The uptick in the 2-methylhopanes in the fossil record is linked to the HpnP-containing Alphaproteobacteria (Hyphomicrobiales). However, I am not seeing a reason why this couldn't alternatively be linked to a shift to more productive (more P rich) and more oxic oceans. The authors stress that 2-methylhopanes can be concentrated when there is an extensive aerobic degradation—that is exactly what we would expect moving into the Neoproterozoic and Phanerozoic.

RESPONSE: The role of aerobic degradation in the observed 2-MHI elevation is difficult to constrain and essentially depends on the nature of aerobic heterotrophic bacteria. Enhanced activity of heterotrophs in general, the majority of which do not produce 2-methylhopanoids, would further reduce the relative abundance of 2-methylhopanoids by dilution with non-methylated hopanoids. However, a relative increase of aerobic HpnP-containing alphaproteobacteria amongst all heterotrophs—whose rise would likely not have been possible without enhanced environmental oxygenation— would have led to higher amounts of 2-methylhopanoid production. Thus, while environmental factors have certainly played an important role in a network of causalities, it is unlikely that enhanced aerobic degradation itself caused the observed signature. The most parsimonious explanation for our data indeed involves the rise of HpnP-containing (heterotrophic) alphaproteobacteria. The role of aerobic degradation is added to the discussion and the conclusion in the main text (see below).

L302-L309 (main text)

The elevated 2-MHI may additionally reflect an increased reworking of newly-available algal biomass in increasingly oxygenated marine environments by heterotrophic alphaproteobacteria. These hypotheses are complementary to the previously proposed relationship between 2-methylhopanoid production and the nitrogen cycle during Cretaceous oceanic anoxic events (Supplementary Text 4)^{23,24}.

2

Helmholtz Centre Potsdam · GFZ German Research Centre for Geosciences · Public Law Foundation
Telegrafenberg · 14473 Potsdam · www.gfz-potsdam.de
Board of Trustees: MinDirig'in Oda Keppler (Chair)
Executive Board: Prof. Dr. Niels Hovius (Spokesman, interim) · Dr. Stefan Schwartz
Payments: Account No. 3093887 · Deutsche Bank · BLZ 120 700 00
IBAN DE8612070000309388700 · BIC Code DEUTDE33HAN · Tax No. 046/149/01166 · VAT DE138407750

HELMHOLTZ

source, provide a link to the Creative Commons license, and indicate if changes were made. In the cases where the authors are anonymous, such as is the case for the reports of anonymous peer reviewers, author attribution should be to 'Anonymous Referee' followed by a clear attribution to the source work. The images or other third party material in this file are included in the article's Creative Commons license, unless indicated otherwise in a credit line to the material. If material is not included in the article's Creative Commons license and your intended use is not permitted by statutory regulation or exceeds the permitted use, you will need to obtain permission directly from the copyright holder. To view a copy of this license, visit <http://creativecommons.org/licenses/by/4.0/>.

In view of enhanced alphaproteobacterial 2-methylhopanoid production going back to the Ediacaran, it is likely that HpnP-containing alphaproteobacteria were responsible for modulating the 2-MHI throughout the Phanerozoic.

L324-L328 (conclusions)

The synchronization between the 2-MHI increase during the Ediacaran and the ecological expansion of HpnP-containing alphaproteobacteria and eukaryotic algae may not be coincidental, involving a vitamin B₁₂-based mutualistic relationship between Alphaproteobacteria and algae and enhanced reworking of algal biomass by Alphaproteobacteria in increasingly oxygenated marine environments.

2) I would like to see more information about the marine ecology of hyphomicrobiales. For instance what portion of the microbial biomass are HpnP-containing Alphaproteobacteria in typical marine environments? Based on this information is reasonable to suggest they would be a the dominate source of 2-methylhopanes?

RESPONSE: The reviewer raises an important point. There is currently little data about the abundance of marine HpnP-containing alphaproteobacteria. However, one previous study found a dominance (>90%) of Hypomicrobiales for the detected metagenomic HpnP in various marine environments (Ricci et al., 2014; see below). Contrarily, another metagenome study in hypersaline Shark Bay revealed the dominance of cyanobacterial HpnP (Garby et al., 2013; see below). Hence, HpnP-containing cyanobacteria may thrive in more specialized environments in modern ecosystems, compared to apparently more versatile HpnP-containing alphaproteobacteria. We added new supplementary sections about the taxonomic diversity of marine 2-methylhopanoid producers in both Alphaproteobacteria and Cyanobacteria (Supplementary Text 5) and the depositional settings of 2-methylhopanoids (Supplementary Text 8). We hope that our study will inspire further quantitative analyses of the sources of 2-methylhopanoids and other important biomarkers in modern environments.

Ricci, J. N. et al. Diverse capacity for 2-methylhopanoid production correlates with a specific ecological niche. *ISME J.* **8**, 675-684 (2014)

3

Helmholtz Centre Potsdam · GFZ German Research Centre for Geosciences · Public Law Foundation
Telegrafenberg · 14473 Potsdam · www.gfz-potsdam.de
Board of Trustees: MinDirig'in Oda Keppler (Chair)
Executive Board: Prof. Dr. Niels Hovius (Spokesman, interim) · Dr. Stefan Schwartz
Payments: Account No. 3093887 · Deutsche Bank · BLZ 120 700 00
IBAN DE8612070000309388700 · BIC Code DEUTDE33HAN · Tax No. 046/149/01166 · VAT DE138407750

HELMHOLTZ

source, provide a link to the Creative Commons license, and indicate if changes were made. In the cases where the authors are anonymous, such as is the case for the reports of anonymous peer reviewers, author attribution should be to 'Anonymous Referee' followed by a clear attribution to the source work. The images or other third party material in this file are included in the article's Creative Commons license, unless indicated otherwise in a credit line to the material. If material is not included in the article's Creative Commons license and your intended use is not permitted by statutory regulation or exceeds the permitted use, you will need to obtain permission directly from the copyright holder. To view a copy of this license, visit <http://creativecommons.org/licenses/by/4.0/>.

Garby, T. J., Walter, M. R., Larkum, A. W. D. & Neilan, B. A. Diversity of cyanobacterial biomarker genes from the stromatolites of Shark Bay, Western Australia. *Environ. Microbiol.* **15**, 1464-1475 (2013).

3) The updated biomarker record makes sense to me – and including it makes the paper much stronger. But it is entering into a fairly debated topic with very limited discussion in the main text, which I do not think will help move forward debate. I think this section needs to be further flushed out. For instance, I would say there needs to be a clear statement about the presence of robust, verifiable ‘typical’ steranes in the Precambrian record.

RESPONSE: We agree that this is a debated topic, however the main text does not leave room for a comprehensive discussion of biomarker syngeneity and hence this aspect must be discussed in the supplementary material. We chose to not discuss steranes in detail, since the main text does not contain any statement about fossil steranes and hence this other biomarker would require a whole new contextual introduction that is not relevant to the core of our discussion. Our biomarker analysis method follows the current standard protocol that was recently established for Proterozoic and Archaean rock samples (French et al., 2015; see below). In Supplementary Text 6 (see below), we now articulate the absence of a detectable amount of steranes, which is consistent with the general absence of typical sterane biomarkers in pre-Cryogenian samples and thus additionally supports the syngeneity of our biomarker data (e.g. Luo et al., 2016; Brocks et al., 2017; see below).

L323-L327 (Supplementary Text 6)

Steranes are below the detection limit in nearly all interior and exterior samples, with an exception of trace amounts in a few exteriors. Hopanes are abundant in both the exterior and the interior of samples for all formations. The abundance of hopanes is nearly identical between the exterior and the interior of all analyzed samples. These observations indicate that the level of surficial contamination for LV09001 drill core is very low

French K. L., Hallmann C., Hope J. M., Schoon P. L., Zumberge J. A., et al. Reappraisal of hydrocarbon biomarkers in Archean rocks. *Proc. Natl. Acad. Sci.*, **112**: 5915-5920 (2015).

4

Helmholtz Centre Potsdam · GFZ German Research Centre for Geosciences · Public Law Foundation
Telegrafenberg · 14473 Potsdam · www.gfz-potsdam.de
Board of Trustees: MinDirig/in Oda Keppler (Chair)
Executive Board: Prof. Dr. Niels Hovius (Spokesman, interim) · Dr. Stefan Schwartzke
Payments: Account No. 3093887 · Deutsche Bank · BLZ 120 700 00
IBAN DE8612070000309388700 · BIC Code DEUTDE33HAN · Tax No. 046/149/01166 · VAT DE138407750

HELMHOLTZ

source, provide a link to the Creative Commons license, and indicate if changes were made. In the cases where the authors are anonymous, such as is the case for the reports of anonymous peer reviewers, author attribution should be to 'Anonymous Referee' followed by a clear attribution to the source work. The images or other third party material in this file are included in the article's Creative Commons license, unless indicated otherwise in a credit line to the material. If material is not included in the article's Creative Commons license and your intended use is not permitted by statutory regulation or exceeds the permitted use, you will need to obtain permission directly from the copyright holder. To view a copy of this license, visit <http://creativecommons.org/licenses/by/4.0/>.

Luo, Q., George, S. C., Xu, Y. & Zhong, N. Organic geochemical characteristics of the Mesoproterozoic Hongshuizhuang Formation from northern China: Implications for thermal maturity and biological sources. *Org. Geochem.* **99**, 23-37 (2016).

Brocks, J. J. et al. The rise of algae in Cryogenian oceans and the emergence of animals. *Nature* **548**, 578-581 (2017).

4) I like the B12 hypothesis, this section is a strength of the paper. But I was left looking for more information. I would suggest taking text to ref

RESPONSE: Due to the word limitation, we expanded Supplementary Text 9 and included additional references therein (particularly, L421-L438).

L421-L438 (Supplementary Text 9)

The wide occurrence of vitamin B₁₂ dependency in green algae reflects an ancient origin of the algal symbiosis with vitamin B₁₂-producing microbes and the accompanied MetE gene loss⁴². However, the distribution of vitamin B₁₂ dependency is not reflected in the phylogeny of green algae. Similarly, microbial taxa that are known to have a symbiotic relationship with algae are not particularly close to each other. Thus, the dependency between green algae and bacteria evolved many times independently in different algal and microbial lineages, including both freshwater and marine species⁴⁵. The symbiosis between green algae and bacteria is not strictly a co-evolutionary relationship, but is characterized by more loose and broad interactions on occasion. For instance, the green alga *Chlamydomonas nivalis* and the alphaproteobacterium *Mesorhizobium loti* (Hypomicrobiales) establish a mutualistic relationship, in exchange of carbon source and vitamin B₁₂, under laboratory culture conditions⁴⁶. *M. loti* is a rhizobacterium that is found in root nodules in native environments⁴⁷ and thus the symbiosis with *C. nivalis* is a non-natural relationship induced by the specific lab condition. Also, *Chlamydomonas reinhardtii* has a mutualistic relationship with multiple bacterial taxa, such as *Rhizobium*, *Shinella*, *Pseudomonas* and *Flavobacterium* (Hypomicrobiales is the most abundant)⁴⁸. Hence, green algae obtain vitamin B₁₂ from a variety of microbial sources that are available at different times and locations and do not require a fixed relationship with specific microbial partners. In turn, vitamin B₁₂-producing bacteria likely obtain carbon sources from various green algae among others.

5

Helmholtz Centre Potsdam · GFZ German Research Centre for Geosciences · Public Law Foundation
Telegrafenberg · 14473 Potsdam · www.gfz-potsdam.de
Board of Trustees: MinDirig'in Oda Keppler (Chair)
Executive Board: Prof. Dr. Niels Hovius (Spokesman, interim) · Dr. Stefan Schwartze
Payments: Account No. 3093887 · Deutsche Bank · BLZ 120 700 00
IBAN DE8612070000309388700 · BIC Code DEUTDE33HAN · Tax No. 046/149/01166 · VAT DE138407750

HELMHOLTZ

source, provide a link to the Creative Commons license, and indicate if changes were made. In the cases where the authors are anonymous, such as is the case for the reports of anonymous peer reviewers, author attribution should be to 'Anonymous Referee' followed by a clear attribution to the source work. The images or other third party material in this file are included in the article's Creative Commons license, unless indicated otherwise in a credit line to the material. If material is not included in the article's Creative Commons license and your intended use is not permitted by statutory regulation or exceeds the permitted use, you will need to obtain permission directly from the copyright holder. To view a copy of this license, visit <http://creativecommons.org/licenses/by/4.0/>.

5) This is a small point but I am not sure I would use symbiosis when talking about algae and B12 producing bacteria. There is some variability in how the term is used – but there are some strict definitions.

RESPONSE: Symbiosis is an umbrella term that encompasses many different modes of interactions between organisms, including endosymbiosis, mutualism, commensalism and parasitism (López-García et al., 2017; see below). Hence, the relationship between green algae and vitamin B₁₂-producing bacteria is symbiotic and the term was in fact used in the original study that first comprehensively described the relationship between green algae and bacteria (Croft et al., 2005; see below). However, mutualism would be a more accurate term for the specific relationship between algae and vitamin B₁₂-producing bacteria. In the revised manuscript, we use both symbiosis and mutualism, depending on the context.

López-García, P., Eme L., Moreira D. Symbiosis in eukaryotic evolution. *J. Theor. Biol.* **404**, 20-33 (2017)

Croft, M., Lawrence, A., Raux-Deery, E., Warren, M. and Smith, A. Algae acquire vitamin B₁₂ through a symbiotic relationship with bacteria. *Nature* **438**, 90-93 (2005)

6) It would be good to see more information about method for the pyrolysis experiments

RESPONSE: We added additional information to the method section.

L437-L450 (main text)

Aliquots (1 mg) of diplopterol (courtesy of P. Adam and P. Schaeffer, University of Strasbourg), 5 α (H)-cholestanol (\geq 95%, Sigma-Aldrich) or cholesterol (\geq 99%, Sigma-Aldrich) were transferred to glass tubes flame-sealed at one end (Duran, 8 mm diameter, 1 mm wall thickness) dissolved in DCM. After DCM evaporation, ~20 mg active carbon (Sigma-Aldrich, DARCO, 100 mesh particle size, CAS: 7440-44-0) were added, tubes were evacuated to \leq 300 mTorr and flame-sealed with a gas torch. After pyrolysis in an oven at 300°C for 24h, tubes were cooled to room temperature, cracked open and the active carbon was transferred with sequential solvent rinses of *n*-hexane, DCM and methanol onto a small silica plug

6

Helmholtz Centre Potsdam · GFZ German Research Centre for Geosciences · Public Law Foundation
Telegrafenberg · 14473 Potsdam · www.gfz-potsdam.de
Board of Trustees: MinDirig'in Oda Keppler (Chair)
Executive Board: Prof. Dr. Niels Hovius (Spokesman, interim) · Dr. Stefan Schwartz
Payments: Account No. 3093887 · Deutsche Bank · BLZ 120 700 00
IBAN DE8612070000309388700 · BIC Code DEUTDE33HAN · Tax No. 046/149/01166 · VAT DE138407750

HELMHOLTZ

source, provide a link to the Creative Commons license, and indicate if changes were made. In the cases where the authors are anonymous, such as is the case for the reports of anonymous peer reviewers, author attribution should be to 'Anonymous Referee' followed by a clear attribution to the source work. The images or other third party material in this file are included in the article's Creative Commons license, unless indicated otherwise in a credit line to the material. If material is not included in the article's Creative Commons license and your intended use is not permitted by statutory regulation or exceeds the permitted use, you will need to obtain permission directly from the copyright holder. To view a copy of this license, visit <http://creativecommons.org/licenses/by/4.0/>.

in a 4 mL SPE glass tube and subsequently extracted with ca. 10 mL *n*-hexane, 10 mL DCM and at least 4 mL methanol. Once solvents were evaporated under a stream of N₂, the pyrolysate was applied onto a silica gel microcolumn (ca. 500 mg in a glasswool-plugged Pasteur-pipette) and the saturated hydrocarbon fraction was eluted with 1.5 dead volumes *n*-hexane followed by GC-MS analysis. Pyrolysates were analyzed on a Thermo Quantum XLS Ultra triple-quadrupole MS coupled to a Thermo Trace GC Ultra fitted with a VF-1 MS column (40 m, 0.15 mm i.d., 0.15 µm film thickness).

Reviewer #2 (Remarks to the Author):

The manuscript by Hoshino & Nettersheim et al. describes extensive efforts to 1) produce an uncontaminated Proterozoic record of 2-methylhopanes (putative markers for cyanobacteria in the rock record) and 2) to re-evaluate the specificity of 2-methylhopanes as cyanobacterial biomarkers using phylogenetic analyses. I had previously reviewed an earlier version of this manuscript for a different journal. Some of the issues raised earlier still persist in the present submission. The authors will thus note some overlap between the two reviews.

RESPONSE: We thank the reviewer's valuable support for reviewing our manuscript twice and for supporting publication of the manuscript. The dataset of the current manuscript is substantially different from the previous version and thus the majority of the manuscript was newly created, rather than performing one-by-one modifications based on previous reviewers' comments. For the current manuscript, the reviewer's comments are answered and incorporated individually.

In the present manuscript, the authors conclude that 2-methylhopanes were present at low levels during the Proterozoic and that before ~750 Ma the most likely source of hopanoids would have been cyanobacteria. The main evidence presented is the inferred vertical inheritance of hpnP in Cyanobacteria including early branching lineages, which contrasts with horizontal inheritance in certain lineages of alphaproteobacteria and thus places the first occurrence of hpnP in cyanobacteria much earlier than in alphaproteobacteria. This work is significant as it aims to resolve the controversy whether 2-methylhopane biosynthesis (via the hpnP gene) originated in cyanobacteria or alphaproteobacteria and which of these groups predominantly contributed to the 2-methylhopane signal in the rock record. The

7

Helmholtz Centre Potsdam · GFZ German Research Centre for Geosciences · Public Law Foundation
Telegrafenberg · 14473 Potsdam · www.gfz-potsdam.de
Board of Trustees: MinDirig'in Oda Keppler (Chair)
Executive Board: Prof. Dr. Niels Hovius (Spokesman, interim) · Dr. Stefan Schwartze
Payments: Account No. 3093887 · Deutsche Bank · BLZ 120 700 00
IBAN DE8612070000309388700 · BIC Code DEUTDE33HAN · Tax No. 046/149/01166 · VAT DE138407750

HELMHOLTZ

source, provide a link to the Creative Commons license, and indicate if changes were made. In the cases where the authors are anonymous, such as is the case for the reports of anonymous peer reviewers, author attribution should be to 'Anonymous Referee' followed by a clear attribution to the source work. The images or other third party material in this file are included in the article's Creative Commons license, unless indicated otherwise in a credit line to the material. If material is not included in the article's Creative Commons license and your intended use is not permitted by statutory regulation or exceeds the permitted use, you will need to obtain permission directly from the copyright holder. To view a copy of this license, visit <http://creativecommons.org/licenses/by/4.0/>.

authors make a case for a predominantly cyanobacterial source prior to 750 Ma and a primarily alphaproteobacterial source after 750 Ma.

I support publication of this work. However, I find two major issues that should be addressed:

RESPONSE: We thank the reviewer for this positive evaluation.

1) The authors focus on the marine sedimentary record of 2-methylhopanes. The problem that 2-methylhopanes are not actually found in extant marine cyanobacteria (Elling et al., 2020; Naafs et al., 2021) needs to be discussed in detail (also in the light of evolution of marine cyanobacteria) as this suggests that the 2-methylhopane geologic record is dependent on depositional setting (e.g., near-shore vs. pelagic) in addition to geologic time (timing of cyanobacterial radiations). I think it is important to discuss this more prominently as this will be a very important aspect for future studies. This discussion is also warranted because the authors exclude Rokubacteria as a source of 2-methylhopanes in marine environments because they have no marine representatives but the same is actually true for hpnP-containing Cyanobacteria. This discussion is also very important for interpreting the assumed shift in 2-methylhopane sources inferred to have occurred around 750-500 Ma. This timeframe was significant for the evolution of cyanobacteria as it coincides with the emergence of extant marine cyanobacterial lineages (Sánchez-Baracaldo, 2015), all of which do not contain 2-methylhopanes. Why was 2-methylhopane biosynthesis lost during that radiation if it was previously a pervasive feature? Importantly, since coastal environments may receive 2-methylhopanoid input from freshwater or tidal environments, I wonder if the shift in 2-methylhopane abundance during that time is biased by different depositional settings for these records?

RESPONSE: We appreciate the reviewer's important insights. As the reviewer suggests, HpnP is mostly absent in cultured marine species in both Cyanobacteria and Alphaproteobacteria. We performed additional molecular analyses. Accordingly, we modified the main text and also added three new supplementary sections (Supplementary Texts 3, 5 and 8) to extensively discuss our current understanding about marine 2-methylhopanoid production and its relationship to the depositional setting. Supplementary Text 3 discusses the HpnP gene loss history in the lineages that contain marine

8

Helmholtz Centre Potsdam · GFZ German Research Centre for Geosciences · Public Law Foundation
Telegrafenberg · 14473 Potsdam · www.gfz-potsdam.de
Board of Trustees: MinDirig'in Oda Keppler (Chair)
Executive Board: Prof. Dr. Niels Hovius (Spokesman, interim) · Dr. Stefan Schwartz
Payments: Account No. 3093887 · Deutsche Bank · BLZ 120 700 00
IBAN DE8612070000309388700 · BIC Code DEUTDE33HAN · Tax No. 046/149/01166 · VAT DE138407750

HELMHOLTZ

source, provide a link to the Creative Commons license, and indicate if changes were made. In the cases where the authors are anonymous, such as is the case for the reports of anonymous peer reviewers, author attribution should be to 'Anonymous Referee' followed by a clear attribution to the source work. The images or other third party material in this file are included in the article's Creative Commons license, unless indicated otherwise in a credit line to the material. If material is not included in the article's Creative Commons license and your intended use is not permitted by statutory regulation or exceeds the permitted use, you will need to obtain permission directly from the copyright holder. To view a copy of this license, visit <http://creativecommons.org/licenses/by/4.0/>.

cyanobacteria. Supplementary Text 5 discusses potential marine 2-methylhopanoid producers in Cyanobacteria and Alphaproteobacteria. Supplementary Text 8 discusses the depositional setting for marine 2-methylhopanoids.

In short, we identified many additional marine species that potentially produce 2-methylhopanoids in both Cyanobacteria and Alphaproteobacteria (but not in Rokubacteria), by exploring genome databases. Hence, 2-methylhopanoid producers are more widely distributed than previously thought. Also, we examined the timing of HpnP gene loss in the lineages that contain marine planktonic species, utilizing recently-published phylogenomic analyses of marine cyanobacteria (Extended Data Fig. 2). We find that HpnP gene loss occurred much earlier than the actual emergence of modern marine planktonic species, possibly in the Paleo- and Mesoproterozoic, although the possibility of the presence of extinct HpnP-containing marine planktonic species is not excluded. Further, we examined the temporal variation of cyanobacterial 2-methylhopanoid production in a large number of very different sedimentary samples throughout Earth's history and argue that the observed 2-MHI trend cannot be explained solely by depositional bias (which should rather favor higher 2-MHI values in the Precambrian), but requires an additional source – *i.e.* Alphaproteobacteria. The possibility of the presence of some depositional bias cannot be fully excluded, as the reviewer suggests, but any depositional bias cannot provide a first-order explanation for the presented observations.

The main text was modified to incorporate and complement the discussions in the supplementary texts (see below for a brief summary).

L208-L210 (main text)

Also, rokubacterial HpnP seems confined to several late-branching clades, although the exact timing of HpnP acquisition is not clear (Extended Data Fig. 3 and Supplementary Figs. 7–9).

L214-L216 (main text)

Also, our genome data reveal that HpnP is in fact distributed much more widely in marine cyanobacteria than previously known²³ (Supplementary Table 5 and Supplementary Text 5 for more details).

L273-L279 (main text)

Ecological shifts throughout the Ediacaran period—diminished abundances of microbial mats^{36,37} and the expansion of HpnP-free cyanobacteria as major nitrogen fixers in pelagic regions (Extended Data Fig. 2)—should have curtailed net cyanobacterial 2-methylhopanoid productions and preservations. This draws particular attention to the unexpected Ediacaran increase of the 2-MHI that probably reflects the emergence of novel, non-cyanobacterial 2-methylhopanoid sources that may have played an important ecological role in Ediacaran oceans (Fig. 4B).

2) In my view, the authors do not adequately discuss prior work. The manuscript builds upon the long-standing controversy surrounding the cyanobacterial vs. alphaproteobacterial origin of 2-methylhopanes. Therefore, the arguments and merits of previous studies should be discussed and refuted individually, not summarily as presently done (particularly Ricci et al., 2015). The authors seem to imply that the majority of cyanobacterial lineages/species/genomes contain SC and hpnP, whereas previous studies have found the opposite to be true (Ricci et al., 2015; Elling et al., 2020; Naafs et al., 2021), see also (see also Fig. 6 in Kusch and Rush, 2022). Particularly, these earlier studies found that hpnP is not present in marine cyanobacteria, even in environments that favored freshwater/brackish lineages of cyanobacteria. Finally, the phylogeny of hpnP derived by the authors differs to some extent from previous studies, some of which placed the cyanobacterial hpnP sequences within alphaproteobacteria. How would these alternative phylogenies impact the conclusions? A more robust discussion on the choice of substitution model/phylogenetic methods and comparison to previous works is warranted here. The lack of this discussion is surprising, given that this formed a major part of a previous version of this manuscript.

RESPONSE: We appreciate the reviewer's important comment. First, we have to clarify that it was not our intention to argue that the majority of Cyanobacteria have HpnP. We infer that HpnP was indeed horizontally transferred multiple times and also experienced gene loss in multiple lineages within the phylum. Therefore, our study is in line with previous studies in this regard (*i.e.* Elling et al., 2020; Naafs et al., 2021). Yet, we also infer that HpnP was present in the common ancestor of Cyanobacteria and this marks the major difference from the previous works, particularly the study by Ricci et al., 2015. We modified the main text to more clarify the sporadic distribution of HpnP in Cyanobacteria and the presence of gene loss (see below). Also, we newly created Supplementary Text 4 to compare our study

10

Helmholtz Centre Potsdam · GFZ German Research Centre for Geosciences · Public Law Foundation
Telegrafenberg · 14473 Potsdam · www.gfz-potsdam.de
Board of Trustees: MinDirig/in Oda Keppler (Chair)
Executive Board: Prof. Dr. Niels Hovius (Spokesman, interim) · Dr. Stefan Schwartze
Payments: Account No. 3093887 · Deutsche Bank · BLZ 120 700 00
IBAN DE86120700000309388700 · BIC Code DEUTDE33HAN · Tax No. 046/149/01166 · VAT DE138407750

HELMHOLTZ

source, provide a link to the Creative Commons license, and indicate if changes were made. In the cases where the authors are anonymous, such as is the case for the reports of anonymous peer reviewers, author attribution should be to 'Anonymous Referee' followed by a clear attribution to the source work. The images or other third party material in this file are included in the article's Creative Commons license, unless indicated otherwise in a credit line to the material. If material is not included in the article's Creative Commons license and your intended use is not permitted by statutory regulation or exceeds the permitted use, you will need to obtain permission directly from the copyright holder. To view a copy of this license, visit <http://creativecommons.org/licenses/by/4.0/>.

in more detail with three important previous studies (Ricci et al., 2015; Elling et al., 2020; Naafs et al., 2021).

L154-L155 (main text)

In contrast to Alphaproteobacteria, the HpnP branching order in Cyanobacteria broadly matches the species tree, despite the overall distribution of HpnP being sporadic at the species-level

L166-L168 (main text)

The absence of HpnP in the majority of Cyanobacteria is attributed to gene loss that seems to have occurred multiple times independently in major lineages

Line comments:

Line 94-96: Occurrence in Actinobacteria described in Naafs et al. 2021 (Fig. 5 and supplement) and Elling et al. (2022, Fig. S2)

RESPONSE: Corrected.

L85-L86 (main text)

This contrasts with only four phyla that were previously known to harbor HpnP-containing species

Line 107-110: The physiological roles of hopanoids have been explored in some detail (e.g., Doughty et al., 2009; Wu et al., 2015).

RESPONSE: The physiological role of methylhopanoids is now discussed in the newly created Supplementary Text 3, as part of the HpnP gene loss discussion in a broader sense. The physiological role of hopanoids is a complicated topic. There is currently no consensus about the general role of hopanoids shared by all hopanoid-producing bacteria, unlike the essentiality and the shared function of steroids in nearly all eukaryotes. As the reviewer suggests, it is known that hopanoids modify the biophysical property of membranes. However, the relationship between hopanoid production and particular cellular processes is still largely unknown, except only for a few examples. Further, those

11

Helmholtz Centre Potsdam · GFZ German Research Centre for Geosciences · Public Law Foundation
Telegrafenberg · 14473 Potsdam · www.gfz-potsdam.de
Board of Trustees: MinDirig'in Oda Keppler (Chair)
Executive Board: Prof. Dr. Niels Hovius (Spokesman, interim) · Dr. Stefan Schwartze
Payments: Account No. 3093887 · Deutsche Bank · BLZ 120 700 00
IBAN DE8612070000309388700 · BIC Code DEUTDE33HAN · Tax No. 046/149/01166 · VAT DE138407750

HELMHOLTZ

source, provide a link to the Creative Commons license, and indicate if changes were made. In the cases where the authors are anonymous, such as is the case for the reports of anonymous peer reviewers, author attribution should be to 'Anonymous Referee' followed by a clear attribution to the source work. The images or other third party material in this file are included in the article's Creative Commons license, unless indicated otherwise in a credit line to the material. If material is not included in the article's Creative Commons license and your intended use is not permitted by statutory regulation or exceeds the permitted use, you will need to obtain permission directly from the copyright holder. To view a copy of this license, visit <http://creativecommons.org/licenses/by/4.0/>.

examples are perhaps taxon-specific and it is unclear if the same function can be extrapolated to other bacterial lineages. The relevant statements in Supplementary Text 3 are shown below.

L89-96 (Supplementary Text 3)

The role of 2-methylhopanoids is thought to be related to stress response, as observed for the elevated production of 2-methylhopanoids by a plant-associated alphaproteobacterium under hypoxic and acidic conditions⁴. Hence, the loss of HpnP may imply the disappearance of a particular environmental stressor for host organisms. For instance, early aerobic organisms may have experienced more frequent and persistent hypoxic conditions and thus HpnP-induced cell protection may have been more advantageous than in later oxygenated worlds. However, it is not clear if this specific observation for terrestrial species can be applicable to organisms in other habitats, in particular marine species.

Line 204-207: This argument was previously made by Ricci et al. (2015), based on earlier work by Summons, which should be credited here.

RESPONSE: We modified the sentence and included Summons et al., 1999.

L187-L191 (main text)

In this framework, the analysis of geological 2-methylhopane records as aerobiosis markers potentially enables us to address the timing of incipient oxygen production before the onset of the GOE at ~2.4 Ga as was originally proposed⁸, regardless of the phylogenetic identity of the biological host, if thermally well-preserved sedimentary sequences should ever be found²².

⁸Summons, R. E., Jahnke, L. L., Hope, J. M. & Logan, G. A. 2-Methylhopanoids as biomarkers for cyanobacterial oxygenic photosynthesis. *Nature* 400, 554-557 (1999).

Line 270-271: Here or elsewhere the depositional setting of the McArthur basin formation should be discussed in the light of possible biases in the 2Me record, since hpnP-containing cyanobacteria are mostly freshwater or benthic/coastal species.

RESPONSE: The depositional setting of 2-methylhopanoids is now discussed in Supplementary Text 8 and the main text, as described above. Considering the depositional setting of the McArthur Basin and inclusion of shallow water facies that likely harboured abundant cyanobacterial mats (that today often contain 2-methylhopane producers), the low 2-MHI values are even more surprising, putting the Ediacaran increase into sharp focus and point towards the emergence of new marine 2-methylhopanoid sources in the late Neoproterozoic.

Line 346-350: This statement is speculative and could be toned down.

L346-L350 in the previous manuscript

“This finding re-establishes the utility of 2-methylhopanes as a biomarker for Cyanobacteria in pre-Ediacaran rocks, thus enabling to measure the importance of HpnP-containing oxygenic Cyanobacteria in the geological past. The 2-MHI reflects the relative abundance of 2-methylhopanoid-producing Cyanobacteria to hopanoid-producing heterotrophic bacteria.”

RESPONSE: As requested by the reviewer, we have expanded the discussion about marine 2-methylhopanoid producers, as described above, yet we still infer that the marine 2-methylhopane record in pre-Ediacaran oceans largely or only reflects 2-methylhopanoids produced by marine cyanobacteria. Hence, pre-Ediacaran 2-methylhopanes are most likely cyanobacterial biomarkers. We still deleted the latter part of the statement because the total hopanoid production on depends not only on heterotrophic bacteria, but also non-methylated hopanoid-producing cyanobacteria. The revised statement is shown below.

L320-L322 (main text)

This finding re-establishes the utility of 2-methylhopanes as a biomarker for Cyanobacteria in pre-Ediacaran rocks, enabling us to measure the importance of HpnP-containing oxygenic cyanobacteria in the geological past.

References

- Doughty, D.M., Hunter, R.C., Summons, R.E., Newman, D.K., 2009. 2-Methylhopanoids are maximally produced in akinetes of *Nostoc punctiforme*: geobiological implications. *Geobiology* 7, 524–532.
- Elling, F.J., Hemingway, J.D., Evans, T.W., Kharbush, J.J., Spieck, E., Summons, R.E., Pearson, A., 2020. Vitamin B12-dependent biosynthesis ties amplified 2-methylhopanoid production during oceanic anoxic events to nitrification. *Proceedings of the National Academy of Sciences* 117, 32996–33004.
- Kusch, S., Rush, D., 2022. Revisiting the precursors of the most abundant natural products on Earth: A look back at 30+ years of bacteriohopanepolyol (BHP) research and ahead to new frontiers. *Organic Geochemistry* 172, 104469.
- Naafs, B.D.A., Bianchini, G., Monteiro, F.M., Sánchez-Baracaldo, P., 2021. The occurrence of 2-methylhopanoids in modern bacteria and the geological record. *Geobiology* n/a. doi:10/gk9j7d
- Ricci, J.N., Michel, A.J., Newman, D.K., 2015. Phylogenetic analysis of HpnP reveals the origin of 2-methylhopanoid production in Alphaproteobacteria. *Geobiology* 13, 267–277.
- Sánchez-Baracaldo, P., 2015. Origin of marine planktonic cyanobacteria. *Scientific Reports* 5, 17418.
- Wu, C.-H., Bialecka-Fornal, M., Newman, D.K., 2015. Methylation at the C-2 position of hopanoids increases rigidity in native bacterial membranes. *eLife* 4, e05663.

Reviewer #3 (Remarks to the Author):

Hoshino et al. study the evolution of the genes underpinning 2-methylhopane biosynthesis in Bacteria. Their analyses aim to address the question of whether these molecules can be used as biomarkers for Cyanobacteria, given the presence of the biosynthetic genes in some other bacterial phyla (Alphaproteobacteria and Acidobacteria). The analyses suggest that these genes were acquired recently by HGT in Alphaproteobacteria, suggesting that the presence of the lipids earlier in the rock record likely reflects the presence of Cyanobacteria --- rehabilitating 2-methylhopanes as biomarkers for early cyanobacterial evolution. The authors then go on to re-appraise the biomarker record for these lipids, including new analyses of the interior and exterior of samples to distinguish original presence from contamination from more recent rocks.

Overall, this is an interesting manuscript, and the combination of phylogenetic analyses and geobiological/biomarker experiments is welcome. My detailed remarks are restricted to the phylogenetic

14

Helmholtz Centre Potsdam · GFZ German Research Centre for Geosciences · Public Law Foundation
Telegrafenberg · 14473 Potsdam · www.gfz-potsdam.de
Board of Trustees: MinDirig/in Oda Keppler (Chair)
Executive Board: Prof. Dr. Niels Hovius (Spokesman, interim) · Dr. Stefan Schwartz
Payments: Account No. 3093887 · Deutsche Bank · BLZ 120 700 00
IBAN DE8612070000309388700 · BIC Code DEUTDE33HAN · Tax No. 046/149/01166 · VAT DE138407750

HELMHOLTZ

source, provide a link to the Creative Commons license, and indicate if changes were made. In the cases where the authors are anonymous, such as is the case for the reports of anonymous peer reviewers, author attribution should be to 'Anonymous Referee' followed by a clear attribution to the source work. The images or other third party material in this file are included in the article's Creative Commons license, unless indicated otherwise in a credit line to the material. If material is not included in the article's Creative Commons license and your intended use is not permitted by statutory regulation or exceeds the permitted use, you will need to obtain permission directly from the copyright holder. To view a copy of this license, visit <http://creativecommons.org/licenses/by/4.0/>.

analyses, which I am (arguably) competent to assess, but the overall impression is that this paper reports an integrated body of work that makes an important contribution to our understanding of the early bacterial (molecular) fossil record. However, there seem to be some important gaps in the analyses that could be plugged fairly straightforwardly, and would help to shore up the authors' conclusions, as I detail below.

RESPONSE: We appreciate the reviewer's positive comments and are very grateful for the important suggestions.

Major comments

- I appreciate that the authors used gene tree-species tree reconciliation to study the evolution of the key HpnP gene in Cyanobacteria. However, I have a couple of questions about these analyses.

First, why was the gene tree reconciled against a species tree that only contained Cyanobacteria that have HpnP? It seems like this analysis will greatly under-estimate the number of losses. For example, the evidence for the gene being present ancestrally in Cyanobacteria would be stronger if the gene is broadly, rather than sporadically, distributed in the clade.

RESPONSE: It is already well known that HpnP is sporadically distributed in Cyanobacteria and this rendered the evolutionary origin of cyanobacterial HpnP a highly controversial topic in the last decade. The distribution of HpnP is complex at the species level and the accurate quantification of gene loss throughout cyanobacterial evolution is not the main focus of our study. Instead, our focus is on addressing the ancestry of HpnP in the common ancestor of Cyanobacteria and the re-interpretation of 2-methylhopanes in the geological record. Our analyses suggest that the gene is taxonomically widely distributed, while it is quantitatively sporadically distributed. Thus, tree reconciliation was performed not to count the HpnP gene loss, but to determine the vertical inheritance of HpnP, which was negated by a previous study that utilized a smaller dataset, compared to ours. Notung results show that the HpnP tree broadly recapitulates the species tree and support the presence of HpnP in the common ancestor of Cyanobacteria. In the revised manuscript, these points are clarified (see below). Also, HpnP gene loss is

discussed in the newly created Supplementary Text 3, in light of the evolution of marine planktonic cyanobacteria, which play critical roles in the modern nitrogen cycle, but do not possess HpnP.

L154-L155 (main text)

In contrast to Alphaproteobacteria, the HpnP branching order in Cyanobacteria broadly matches the species tree, **despite the overall distribution of HpnP being sporadic at the species level**

L166-L168 (main text)

The absence of HpnP in the majority of Cyanobacteria is attributed to gene loss that seems to have occurred multiple times independently in major lineages

On a related note, how does Notung determine the probability (or, I suppose, support) for a gene to be present at the root of the species tree --- for example, how much less parsimonious were scenarios in which the gene was acquired later within Cyanobacteria and then passed around by transfer? Bottoming out these issues is important because - as I understand it - it is the fit between the gene and species tree that the authors are using to date the gene acquisition (by dating the root of the Cyanobacteria in the species tree).

RESPONSE: Notung calculates the number of evolutionary events that are required to reconcile the input trees: duplication, horizontal transfer and gene loss (DTL model). Hence, there is no probability-based support values for the results. Instead, the number of events is regarded as the support (the smaller is the better). In our analyses, the ‘best’ results, requiring the smallest number of evolutionary events, all supported the presence of HpnP in the common ancestor of Cyanobacteria. In contrast, scenarios that explain the presence of HpnP in early-branching species by horizontal gene transfer required larger numbers of such evolutionary events. While this parsimonious approach does not necessarily trace the accurate evolutionary history of a gene, it provides auxiliary quantitative information for phylogenetic analyses.

- On a related note, why not perform the same analysis for Alphaproteobacteria? Presumably the results will be quite different to the cyanobacterial case (i.e. the reconciliation method will map the gene to a

16

Helmholtz Centre Potsdam · GFZ German Research Centre for Geosciences · Public Law Foundation
Telegrafenberg · 14473 Potsdam · www.gfz-potsdam.de
Board of Trustees: MinDirig/in Oda Keppler (Chair)
Executive Board: Prof. Dr. Niels Hovius (Spokesman, interim) · Dr. Stefan Schwartze
Payments: Account No. 3093887 · Deutsche Bank · BLZ 120 700 00
IBAN DE8612070000309388700 · BIC Code DEUTDE33HAN · Tax No. 046/149/01166 · VAT DE138407750

HELMHOLTZ

source, provide a link to the Creative Commons license, and indicate if changes were made. In the cases where the authors are anonymous, such as is the case for the reports of anonymous peer reviewers, author attribution should be to 'Anonymous Referee' followed by a clear attribution to the source work. The images or other third party material in this file are included in the article's Creative Commons license, unless indicated otherwise in a credit line to the material. If material is not included in the article's Creative Commons license and your intended use is not permitted by statutory regulation or exceeds the permitted use, you will need to obtain permission directly from the copyright holder. To view a copy of this license, visit <http://creativecommons.org/licenses/by/4.0/>.

much shallower node on the dated species tree), providing an empirical test of the hypothesis that the gene traces deeper in Cyanobacteria than in Alphaproteobacteria.

RESPONSE: HpnP is distributed nearly exclusively in a few discrete families within the late-branching Hypomicrobiales and those alphaproteobacterial HpnP appeared too late in Earth's history to account for the majority of 2-methylhopane biomarkers, in particular pre-Ediacaran biomarker records. Therefore, whether HpnP was spread within those families vertically or horizontally (after 750 Ma) does not change the results of our study.

- The HpnP gene tree suggests that the gene may also be rather ancient in Rokubacteria (at least, there is a substantial clade of homologues in that phylum). How does the gene tree here compare to the species tree? The authors mention that they think it unlikely that these bacteria had much of a role in generating the lipids early in evolution, but do not provide a great deal of evidence for this. It would be worthwhile performing the same analysis here as for Cyanobacteria (and, in my view, Alphaproteobacteria) --- is the gene transferred a lot in Rokubacteria, or does it date to their common ancestor? I appreciate that the authors have done quite a lot of work, and may not wish to (for example) estimate the age of the Rokubacterial ancestor using molecular clocks, but at least testing whether the gene might have been present at their root or not seems quite important to the main question being tested here. For example, if the gene actually seems to be recently acquired (as in Alphaproteobacteria), the authors' case that only Cyanobacteria are likely to have contributed to the biomarker record early on would be greatly strengthened. As it stands, the question of Rokubacteria seems to be something of a loose thread.

RESPONSE: We appreciate the reviewer's important insights and fully agree with the necessity to perform additional molecular analyses for Rokubacteria. Unfortunately, this clade is still poorly sampled and all of the HpnP genes come from metagenomic data. Thus, there is just not enough information available to construct a reliable species tree. Hence, upon the reviewer's request we performed only a simplified analysis and the results need to be regarded as tentative. We created a rokubacterial species tree, utilizing three ribosomal proteins L2, L3 and L4 (Extended Data Fig. 3 and Supplementary Fig. 7). Intriguingly, HpnP was found only in several late-branching clades. In contrast, SC is widespread in the phylum and the common ancestor of Rokubacteria likely had the ability to produce non-methylated

17

Helmholtz Centre Potsdam · GFZ German Research Centre for Geosciences · Public Law Foundation
Telegrafenberg · 14473 Potsdam · www.gfz-potsdam.de
Board of Trustees: MinDirig'in Oda Keppler (Chair)
Executive Board: Prof. Dr. Niels Hovius (Spokesman, interim) · Dr. Stefan Schwartze
Payments: Account No. 3093887 · Deutsche Bank · BLZ 120 700 00
IBAN DE8612070000309388700 · BIC Code DEUTDE33HAN · Tax No. 046/149/01166 · VAT DE138407750

HELMHOLTZ

source, provide a link to the Creative Commons license, and indicate if changes were made. In the cases where the authors are anonymous, such as is the case for the reports of anonymous peer reviewers, author attribution should be to 'Anonymous Referee' followed by a clear attribution to the source work. The images or other third party material in this file are included in the article's Creative Commons license, unless indicated otherwise in a credit line to the material. If material is not included in the article's Creative Commons license and your intended use is not permitted by statutory regulation or exceeds the permitted use, you will need to obtain permission directly from the copyright holder. To view a copy of this license, visit <http://creativecommons.org/licenses/by/4.0/>.

hopanoids. Notung analyses suggest the HpnP tree is broadly in agreement with the created species tree, while the results also suggest the presence of horizontal gene transfer and gene loss in individual rokubacterial clades, similar to Cyanobacteria and Alphaproteobacteria (Supplementary Figs. 8 and 9). Molecular clock analyses were not performed, due to the potential low data quality. These results are briefly described in the revised main text (see below).

L208-L210 (main text)

Also, rokubacterial HpnP seems confined to several late-branching clades, although the exact timing of HpnP acquisition is not clear (Extended Data Fig. 3 and Supplementary Figs. 7–9).

Minor comments

- "transferred vertically" - avoid, maybe "inherited vertically".

RESPONSE: Corrected. Similarly, “vertical transfer” in Fig. 3 was also corrected.

L135-L136 (main text)

HpnP was **inherited** vertically in crown-group Cyanobacteria

- line 156: Eremiobacterota misspelled?

RESPONSE: Corrected.

- I would modify the Figure 4 legend slightly to indicate that both the topology and the inferred species tree dates are based on the recent study cited.

RESPONSE: Modified.

L485 (Fig. 4 figure caption)

Both the tree topology and the node dates were adapted from a recent phylogenomic study

Reviewer #4 (Remarks to the Author):

Helmholtz Centre Potsdam · GFZ German Research Centre for Geosciences · Public Law Foundation
Telegrafenberg · 14473 Potsdam · www.gfz-potsdam.de
Board of Trustees: MinDirig/in Oda Keppler (Chair)
Executive Board: Prof. Dr. Niels Hovius (Spokesman, interim) · Dr. Stefan Schwartz
Payments: Account No. 3093887 · Deutsche Bank · BLZ 120 700 00
IBAN DE8612070000309388700 · BIC Code DEUTDE33HAN · Tax No. 046/149/01166 · VAT DE138407750

HELMHOLTZ

18

source, provide a link to the Creative Commons license, and indicate if changes were made. In the cases where the authors are anonymous, such as is the case for the reports of anonymous peer reviewers, author attribution should be to 'Anonymous Referee' followed by a clear attribution to the source work. The images or other third party material in this file are included in the article's Creative Commons license, unless indicated otherwise in a credit line to the material. If material is not included in the article's Creative Commons license and your intended use is not permitted by statutory regulation or exceeds the permitted use, you will need to obtain permission directly from the copyright holder. To view a copy of this license, visit <http://creativecommons.org/licenses/by/4.0/>.

General comments,

This is a nice work presented by Hoshino et al., which puts a genetic constrain on the interpretation of fossil 2Me-hopanoids in geological record. The BLAST search of HpnP gene in additional genomes, phylogenetic analysis of identified homologs, and the molecular clock estimates all yield valuable information for understanding the evolution history of 2Me-hopanoids producing microbes. And, the revised 2-MHI dataset that eliminates bias of contamination is a great contribution to the field.

My only critique lies with the authors' interpretation on the high 2-MHI of Phanerozoic. Naafs et al., (2021, *Geobiology*) conducted very similar works, including phylogenetic analyses of HpnP, biological production of 2Me-hopanoids and the geological record of 2-MHI and related the Phanerozoic 2Me-hopanoids distribution to marine nitrogen cycle. I don't know why Naafs' work is not mentioned in this study. It is nice that Hoshino has proposed a potential connection between the rise of algae and B12 producing alphaproteobacteria in the Phanerozoic, but the coincidence of extreme high 2-MHI with marine anoxic events and the ecological role of alphaproteobacterial in marine nitrogen cycle can be further discussed. Another recent publication, Garby et al. (2022, *EMR*), reported the dominant 2Me-hopanoids contribution by cyanobacteria over alphaproteobacteria under harsh/extreme conditions. Additional discussion on the role of environmental stress influencing the geological distribution of 2Me-hopanoids can also enrich this paper.

RESPONSE: We thank the reviewer for the positive comments and also for important insights. Naafs et al., 2021 is in fact cited in our manuscript, but the work was not explicitly discussed in the previous version. Due to the word limitation, we newly created Supplementary Text 4 to discuss the similarity and also the difference between our present study and previous works, including Naafs et al., 2021. Our hypothesis about algal-microbial relationship is not exclusive to the previously-proposed relationship between the 2-MHI and the nitrogen cycle. Our hypothesis serves more as the baseline to elucidate the overall elevation of 2-methylhopanoid abundance in post-Cryogenian oceans. The occasionally-observed occurrences of extremely high 2-MHI values (such as during oceanic anoxic events) require additional mechanisms, including hypotheses proposed by Elling et al., 2020 and Naafs et al., 2021, but our results support the inference of likely alphaproteobacterial contributions, yet point towards the

19

Helmholtz Centre Potsdam · GFZ German Research Centre for Geosciences · Public Law Foundation
Telegrafenberg · 14473 Potsdam · www.gfz-potsdam.de
Board of Trustees: MinDirig'in Oda Keppler (Chair)
Executive Board: Prof. Dr. Niels Hovius (Spokesman, interim) · Dr. Stefan Schwartze
Payments: Account No. 3093887 · Deutsche Bank · BLZ 120 700 00
IBAN DE8612070000309388700 · BIC Code DEUTDE33HAN · Tax No. 046/149/01166 · VAT DE138407750

HELMHOLTZ

source, provide a link to the Creative Commons license, and indicate if changes were made. In the cases where the authors are anonymous, such as is the case for the reports of anonymous peer reviewers, author attribution should be to 'Anonymous Referee' followed by a clear attribution to the source work. The images or other third party material in this file are included in the article's Creative Commons license, unless indicated otherwise in a credit line to the material. If material is not included in the article's Creative Commons license and your intended use is not permitted by statutory regulation or exceeds the permitted use, you will need to obtain permission directly from the copyright holder. To view a copy of this license, visit <http://creativecommons.org/licenses/by/4.0/>.

possibility of other (earlier diverging) alphaproteobacterial sources not explicitly discussed in the earlier papers. As proposed by the reviewer, we added a short discussion to the main text (see below).

L304-L309 (main text)

These hypotheses are complementary to the previously proposed relationship between 2-methylhopanoid production and the nitrogen cycle during Cretaceous oceanic anoxic events (Supplementary Text 4)^{23,24}. In view of enhanced alphaproteobacterial 2-methylhopane production going back to the Ediacaran, it is likely that HpnP-containing alphaproteobacteria were responsible for modulating the 2-MHI throughout the Phanerozoic.

While Garby et al., 2022 presents an important observation of cyanobacterial HpnP in harsh terrestrial settings, the results do not directly relate to our marine 2-methylhopane records. Also, the study did not measure 2-methylhopanoids themselves and thus does not provide information about the actual 2-methylhopanoid production rate, which was extensively discussed by Elling et al., 2020 and Naafs et al., 2021. Hence, follow-up studies are clearly needed to accurately quantify 2-methylhopanoid production in such settings by Cyanobacteria and Alphaproteobacteria, respectively.

Naafs, B.D.A., Bianchini, G., Monteiro, F.M., Sánchez-Baracaldo, P., 2021. The occurrence of 2-methylhopanoids in modern bacteria and the geological record. *Geobiology* 20, 41–59.

Garby, T. J., Jordan, M., Timms, V., Walter, M. R., & Neilan, B. A. (2022). 2-Methylhopanoids in geographically distinct, arid biological soil crusts are primarily cyanobacterial in origin. *Environmental Microbiology Reports*, 14(1), 164-169.

Specific comments,

Line 39, typo, change 'form' to 'from'

RESPONSE: Corrected.

Line 93-94, only 27 phyla containing SC and 10 of HpnP are shown in Figure 1

20

Helmholtz Centre Potsdam · GFZ German Research Centre for Geosciences · Public Law Foundation
Telegrafenberg · 14473 Potsdam · www.gfz-potsdam.de
Board of Trustees: MinDirig'in Oda Keppler (Chair)
Executive Board: Prof. Dr. Niels Hovius (Spokesman, interim) · Dr. Stefan Schwartz
Payments: Account No. 3093887 · Deutsche Bank · BLZ 120 700 00
IBAN DE8612070000309388700 · BIC Code DEUTDE33HAN · Tax No. 046/149/01166 · VAT DE138407750

HELMHOLTZ

source, provide a link to the Creative Commons license, and indicate if changes were made. In the cases where the authors are anonymous, such as is the case for the reports of anonymous peer reviewers, author attribution should be to 'Anonymous Referee' followed by a clear attribution to the source work. The images or other third party material in this file are included in the article's Creative Commons license, unless indicated otherwise in a credit line to the material. If material is not included in the article's Creative Commons license and your intended use is not permitted by statutory regulation or exceeds the permitted use, you will need to obtain permission directly from the copyright holder. To view a copy of this license, visit <http://creativecommons.org/licenses/by/4.0/>.

RESPONSE: As explained in the figure caption, individual proteobacterial classes are treated as different phyla, due to their substantial taxonomic diversity as well as the particular importance of Alphaproteobacteria, equal to Cyanobacteria, in this manuscript.

Line 104-105, it is not clear to me how this is inferred.

RESPONSE: The paragraph was modified and the confusing sentence was deleted.

L90-L93 (main text)

Despite the sporadic occurrence of SC amongst bacteria, a recent comprehensive phylogenetic study suggests that SC was present in the individual common ancestors of three HpnP-containing phyla (*i.e.* Rokubacteria, Alphaproteobacteria and Cyanobacteria).

Line 131-133, Any studies have reported or discussed fossil hopanoids recycling by microbes?

RESPONSE: We appreciate the reviewer's important suggestion. As far as we are aware of, there is no direct observation of hopanoid recycling by microbes. However, the recalcitrant nature of the hopanoid carbon structure enable hopanoids to serve as biomarkers in the geological record and thus it is feasible that a certain fraction of hopanoids remain intact for a certain amount of time and are recycled by microbes. This potentially has an important ecological implication, but is not directly related to the main focus of this manuscript and thus is not particularly discussed in the current manuscript.

Line 181, please provide the estimated percentage of gene loss.

RESPONSE: The dataset includes only HpnP-containing species and thus the HpnP gene loss was not estimated. HpnP gene loss is widespread across all major lineages and in some cases species-specific. The accurate quantification of gene loss events in the entire cyanobacterial phylum requires detailed phylogenetic analyses of all major cyanobacterial species, which is beyond the scope of this present study. We modified the main text to avoid confusion about HGT and gene loss estimation (see below). We also added HpnP gene loss discussions in more details in Supplementary Text 3.

21

Helmholtz Centre Potsdam · GFZ German Research Centre for Geosciences · Public Law Foundation
Telegrafenberg · 14473 Potsdam · www.gfz-potsdam.de
Board of Trustees: MinDirig/in Oda Keppler (Chair)
Executive Board: Prof. Dr. Niels Hovius (Spokesman, interim) · Dr. Stefan Schwartz
Payments: Account No. 3093887 · Deutsche Bank · BLZ 120 700 00
IBAN DE8612070000309388700 · BIC Code DEUTDE33HAN · Tax No. 046/149/01166 · VAT DE138407750

HELMHOLTZ

source, provide a link to the Creative Commons license, and indicate if changes were made. In the cases where the authors are anonymous, such as is the case for the reports of anonymous peer reviewers, author attribution should be to 'Anonymous Referee' followed by a clear attribution to the source work. The images or other third party material in this file are included in the article's Creative Commons license, unless indicated otherwise in a credit line to the material. If material is not included in the article's Creative Commons license and your intended use is not permitted by statutory regulation or exceeds the permitted use, you will need to obtain permission directly from the copyright holder. To view a copy of this license, visit <http://creativecommons.org/licenses/by/4.0/>.

L162-L164 (main text)

The frequency of HGT is estimated to be no higher than ~15% (13 out of 86 branches) within HpnP-containing cyanobacterial species.

L166-L169 (main text)

The absence of HpnP in the majority of Cyanobacteria is attributed to gene loss that seems to have occurred multiple times independently in major lineages (Extended Data Fig. 2 and Supplementary Text 3 for detailed discussions).

Line 207-214, these two sentences do not fit very well to the context of this subsection. These are discussions of general biomarker preservation and detection, nothing specific of 2Me-Hopane. GC-APLI-TOF was only reported to be highly sensitive for polyaromatic hydrocarbons detection, probably because they are preferentially ionized than others.

RESPONSE: We agree and have drastically shortened the relevant section.

L187-L191 (main text)

In this framework, the analysis of geological 2-methylhopane records as aerobiosis markers potentially enables us to address the timing of incipient oxygen production before the onset of the GOE at ~2.4 Ga as was originally proposed⁸, regardless of the phylogenetic identity of the biological host, if thermally well-preserved sedimentary sequences should ever be found²².

Line 230, delete “and possibly Rokubacteria”, because there is no data to support this. The following discussion on their unlikely contribution to fossil 2-Me hopane is good enough.

RESPONSE: The phrase was deleted.

Line 247, please refer to Fig. 4A in the bracket.

RESPONSE: Modified.

L225 (main text)

(No. 5 in Fig. 4A; Nodes a, b and c)

The calculation of these node bars (a, b and c) is very important for this work. However, the nodes estimation is neither well described in text nor the caption of Fig. 4. It is referred to Fig. S6, but no such information provided in Fig. S6.

RESPONSE: Both tree topology and molecular clock data are derived from a recently published different study (not our own work) and thus the analysis method was not explained in details in our manuscript. More information was added to Supplementary Fig. 10 caption as shown below, but further details are referred to the original study. Figure 4 caption was also modified to clarify the identical source for tree topology and dating data (see below).

L646-L651 (Supplementary Fig. 10 caption)

Published dated phylogeny of Cyanobacteria and Alphaproteobacteria¹⁴ and the superimposition of the simplified tree that is used in Fig. 4A. The superimposed three red triangles represent Beijerinckiaceae (top), Methylobacteriaceae (middle) and Nitrobacteraceae (bottom). In Wang et al., 2020, relaxed molecular clock analyses were performed utilizing this concatenated tree based on the amino acid sequences of 25 universally conserved single-copy genes. Further details are referred to the original study.

L485-L486 (Figure 4 caption)

Both the tree topology and the node dates were adapted from a recent phylogenomic study (Supplementary Fig. 10)³².

Line 253, those triangles are in red not brown in Fig. 4A.

RESPONSE: Corrected.

L230 (main text)

Fig. 4A; three triangles in red

23

Helmholtz Centre Potsdam · GFZ German Research Centre for Geosciences · Public Law Foundation
Telegrafenberg · 14473 Potsdam · www.gfz-potsdam.de
Board of Trustees: MinDirig/in Oda Keppler (Chair)
Executive Board: Prof. Dr. Niels Hovius (Spokesman, interim) · Dr. Stefan Schwartz
Payments: Account No. 3093887 · Deutsche Bank · BLZ 120 700 00
IBAN DE8612070000309388700 · BIC Code DEUTDE33HAN · Tax No. 046/149/01166 · VAT DE138407750

HELMHOLTZ

source, provide a link to the Creative Commons license, and indicate if changes were made. In the cases where the authors are anonymous, such as is the case for the reports of anonymous peer reviewers, author attribution should be to 'Anonymous Referee' followed by a clear attribution to the source work. The images or other third party material in this file are included in the article's Creative Commons license, unless indicated otherwise in a credit line to the material. If material is not included in the article's Creative Commons license and your intended use is not permitted by statutory regulation or exceeds the permitted use, you will need to obtain permission directly from the copyright holder. To view a copy of this license, visit <http://creativecommons.org/licenses/by/4.0/>.

Line 636, do nodes a, b and c represent Be, Ni and Me, respectively?

RESPONSE: The information was added to the figure caption (see below).

L489-L491 (Fig. 4 figure caption)

5. divergence of HpnP-containing Hyphomicrobiales families – **Beijerinckiaceae (Node a), Methylobacteriaceae (Node b) and Nitrobacteraceae (Node c).**

Last sentence of Supplementary Text 3

“, which do not possess a functional group in the A-ring,”

RESPONSE: Corrected.

Thank you for your consideration of this manuscript.

Sincerely,

Yosuke Hoshino

Decision Letter, second revision:

28th July 2023

Dear Dr. Hoshino,

Thank you for submitting your revised manuscript "Genetics re-establish the utility of 2-methylhopanes as cyanobacterial biomarkers before 750 million years ago" (NATECOLEVOL-220316014B). It has now been seen again by the original reviewers and their comments are below. The reviewers find that the paper has improved in revision, and therefore we'll be happy in principle to publish it in Nature Ecology & Evolution, pending minor revisions to satisfy the reviewers' final requests and to comply with our editorial and formatting guidelines.

[REDACTED]

Reviewer #1 wasn't able to provide a full report but noted that they had read the revision and felt it was much improved.

Reviewer #2 (Remarks to the Author):

The revised manuscript is much improved and I have only minor additional suggestions, described below.

L273-276: It is unclear if the authors are referring to the expansion of the ProSyn clade or other cyanobacteria? Not all or even most marine planktonic cyanobacteria are nitrogen fixers, especially in the SynPro clade. The sentence should be revised accordingly.

Supplementary Fig. 10: Ref 14 in the caption seems to be referring to the wrong publication?

Reviewer #3 (Remarks to the Author):

39The authors have done a good job in responding to my comments, and those of the other reviewers, with text edits and new analyses.

With regard to my comments on the previous version, the only comment that is perhaps not fully addressed is the question of whether Rokubacteria might have been able to make 2-methylhopanes prior to 750Ma, and therefore might have made some contribution to that early record. However, the additional analyses (species tree and species tree-gene tree reconciliation) do help to bring some clarity by determining the point on the species tree at which Rokubacteria acquired the required gene. The analysis leaves open an interesting future question, which would be to date that particular node in the rokubacterial species tree (node 6 in Supp Fig 9) --- in particular, to determine if it is older than 750Ma --- but that is perhaps beyond the scope of this already broad-ranging work.

Our ref: NATECOLEVOL-220316014B

11th August 2023

Dear Dr. Hoshino,

Thank you for your patience as we've prepared the guidelines for final submission of your Nature Ecology & Evolution manuscript, "Genetics re-establish the utility of 2-methylhopanes as cyanobacterial biomarkers before 750 million years ago" (NATECOLEVOL-220316014B). Please carefully follow the step-by-step instructions provided in the attached file, and add a response in each row of the table to indicate the changes that you have made. Please also check and comment on any additional marked-up edits we have proposed within the text. Ensuring that each point is addressed will help to ensure that your revised manuscript can be swiftly handed over to our production team.

****We would like to start working on your revised paper, with all of the requested files and forms, as soon as possible (preferably within two weeks). Please get in contact with us immediately if you anticipate it taking more than two weeks to submit these revised files.****

If you have not done so already, please alert us to any related manuscripts from your group that are under consideration or in press at other journals, or are being written up for submission to other

40journals (see: <https://www.nature.com/nature-research/editorial-policies/plagiarism#policy-on-duplicate-publication> for details).

In recognition of the time and expertise our reviewers provide to Nature Ecology & Evolution's editorial process, we would like to formally acknowledge their contribution to the external peer review of your manuscript entitled "Genetics re-establish the utility of 2-methylhopanes as cyanobacterial biomarkers before 750 million years ago". For those reviewers who give their assent, we will be publishing their names alongside the published article.

Nature Ecology & Evolution offers a Transparent Peer Review option for new original research manuscripts submitted after December 1st, 2019. As part of this initiative, we encourage our authors to support increased transparency into the peer review process by agreeing to have the reviewer comments, author rebuttal letters, and editorial decision letters published as a Supplementary item. When you submit your final files please clearly state in your cover letter whether or not you would like to participate in this initiative. Please note that failure to state your preference will result in delays in accepting your manuscript for publication.

Cover suggestions

As you prepare your final files we encourage you to consider whether you have any images or illustrations that may be appropriate for use on the cover of Nature Ecology & Evolution.

Nature Ecology & Evolution has now transitioned to a unified Rights Collection system which will allow our Author Services team to quickly and easily collect the rights and permissions required to publish your work. Approximately 10 days after your paper is formally accepted, you will receive an email in providing you with a link to complete the grant of rights. If your paper is eligible for Open Access, our Author Services team will also be in touch regarding any additional information that may be required to arrange payment for your article.

Please note that Nature Ecology & Evolution is a Transformative Journal (TJ). Authors may publish their research with us through the traditional subscription access route or make their paper

41immediately open access through payment of an article-processing charge (APC). Authors will not be required to make a final decision about access to their article until it has been accepted. [Find out more about Transformative Journals](https://www.springernature.com/gp/open-research/transformative-journals)

Authors may need to take specific actions to achieve [compliance with funder and institutional open access mandates](https://www.springernature.com/gp/open-research/funding/policy-compliance-faqs). If your research is supported by a funder that requires immediate open access (e.g. according to [Plan S principles](https://www.springernature.com/gp/open-research/plan-s-compliance)) then you should select the gold OA route, and we will direct you to the compliant route where possible. For authors selecting the subscription publication route, the journal's standard licensing terms will need to be accepted, including [self-archiving and license to publish](https://www.nature.com/nature-portfolio/editorial-policies/self-archiving-and-license-to-publish). Those licensing terms will supersede any other terms that the author or any third party may assert apply to any version of the manuscript.

[REDACTED]

[REDACTED]

Reviewer #2:

Remarks to the Author:

The revised manuscript is much improved and I have only minor additional suggestions, described below.

L273-276: It is unclear if the authors are referring to the expansion of the ProSyn clade or other cyanobacteria? Not all or even most marine planktonic cyanobacteria are nitrogen fixers, especially in the SynPro clade. The sentence should be revised accordingly.

Supplementary Fig. 10: Ref 14 in the caption seems to be referring to the wrong publication?

Reviewer #3:

Remarks to the Author:

The authors have done a good job in responding to my comments, and those of the other reviewers, with text edits and new analyses.

With regard to my comments on the previous version, the only comment that is perhaps not fully addressed is the question of whether Rokubacteria might have been able to make 2-methylhopanes prior to 750Ma, and therefore might have made some contribution to that early record. However, the additional analyses (species tree and species tree-gene tree reconciliation) do help to bring some clarity by determining the point on the species tree at which Rokubacteria acquired the required gene. The analysis leaves open an interesting future question, which would be to date that particular node in the rokubacterial species tree (node 6 in Supp Fig 9) --- in particular, to determine if it is older than 750Ma --- but that is perhaps beyond the scope of this already broad-ranging work.

Reviewer #4:

None

Author Rebuttal, second revision:

Reviewer #2 (Remarks to the Author):

The revised manuscript is much improved and I have only minor additional suggestions, described below.

L273-276: It is unclear if the authors are referring to the expansion of the ProSyn clade or other cyanobacteria? Not all or even most marine planktonic cyanobacteria are nitrogen fixers, especially in the SynPro clade. The sentence should be revised accordingly.

RESPONSE: The sentence was modified as below (yellow highlighted).

L275-L276

the expansion of HpnP-free cyanobacteria as major **primary producers** in pelagic regions

Supplementary Fig. 10: Ref 14 in the caption seems to be referring to the wrong publication?

RESPONSE: The reviewer is referring to the reference for Supplementary Table 7, not for Supplementary Fig. 10. We modified the reference list to prevent confusions and all references are now in Supplementary Information references.

Reviewer #3 (Remarks to the Author):

The authors have done a good job in responding to my comments, and those of the other reviewers, with text edits and new analyses.

With regard to my comments on the previous version, the only comment that is perhaps not fully addressed is the question of whether Rokubacteria might have been able to make 2-methylhopanes prior to 750Ma, and therefore might have made some contribution to that early record. However, the additional analyses (species tree and species tree-gene tree reconciliation) do help to bring some clarity by determining the point on the species tree at which Rokubacteria acquired the required gene. The analysis leaves open an interesting future question, which would be to date that particular node in the rokubacterial species tree (node 6 in Supp Fig 9) --- in particular, to determine if it is older than 750Ma --- but that is perhaps beyond the scope of this already broad-ranging work.

RESPONSE: We decided to leave it an open question when Rokubacteria acquired HpnP because the species relationship of Rokubacteria with other bacteria is not well constrained and there is also no available calibration point within the phylum, unlike Cyanobacteria and Alphaproteobacteria. Hence, any molecular clock analyses at this stage are potentially inaccurate. It is also dependent on future studies if there are any marine rokubacterial species that have HpnP.

Final Decision Letter:

44